# DISCOVERING THE QUESTION-CRITICAL MOMENTS: TOWARDS BUILDING EVENT-AWARE MULTI-MODAL LARGE LANGUAGE MODELS FOR COMPLEX VIDEO QUESTION ANSWERING

## ABSTRACT

Recently, Multi-modal Large Language Models (MLLM) have demonstrated impressive capabilities in image-language reasoning tasks like Image Question Answering. However, naively transferring them to complex Video Question Answering (VideoQA) tasks suffers from unsatisfactory causal-temporal reasoning capabilities. Existing methods simply concatenate the uniformly sampled frame representations to obtain the video representation, which either results in a quite large number of visual tokens and is thus resource-demanding, or is distracted by the redundancy of question-irrelevant contents. In light of this, we introduce *E-STR*, extending MLLM to be *E*vent-aware for *S*patial-*T*emporal *R*easoning in complex VideoQA tasks. Specifically, we propose a differentiable question-critical keyframes retriever to adaptively select the question-critical moments in the video serving as the key event for spatial-temporal reasoning, and a general context encoder to encode the unselected parts for preserving the general contexts of the video. To facilitate the acquisition of spatial-temporal representations, we also incorporate lightweight adapters within the frozen image encoder. Extensive experiments on three large-scale benchmarks, including NExT-QA, Causal-VidQA, and STAR, all of which are notable for complex causal-temporal reasoning within long videos containing multiple objects and events, show that our method achieves better performance than existing state-of-the-art methods.

## 1 INTRODUCTION

With the recent development of large-scale self-supervised learning, Large Language Models (LLM) have shown powerful reasoning abilities both in daily life and academic benchmarks (Touvron et al., 2023; Brown et al., 2020). To further extend their capabilities, researchers have endeavored to equip LLMs with the ability to understand multi-modal representations such as images and languages, contributing to the Multi-modal Large Language Models (MLLM). Various MLLM variants (Li et al., 2023a; Liu et al., 2023; Ye et al., 2023) share very similar model architecture (Figure 2 (a)) and training objectives (next token prediction (Brown et al., 2020)), having undergone large-scale image-text pre-training, and achieve significant results on a broad range of image-language reasoning tasks like Image Question Answering (Schwenk et al., 2022; Marino et al., 2019; Goyal et al., 2017).

While MLLMs have shown strong image-language reasoning abilities, the exploration of their video-language reasoning abilities is still in an early stage. To solve tasks like simple VideoQA, existing models (Dai et al., 2023; Alayrac et al., 2022; Zhang et al., 2023) uniformly sample sparse frames (e.g., 4 frames for each video) and simply concatenate them (Figure 2 (b)) to represent the entire video, and have achieved fair performance. However, this strategy falls short in complex VideoQA tasks. In this paper, we exploit the image-text pre-trained MLLM for complex VideoQA tasks.

In contrast to straightforward questions featuring perception tasks like detecting objects and their attributes within short videos ($\sim$10s) in simple VideoQA tasks (Xu et al., 2017; Yu et al., 2019), as shown in Figure 1 (a), complex VideoQA tasks require models to engage in intricate causal and temporal inference in long videos with rich visual content and complicated questions, seamlessly

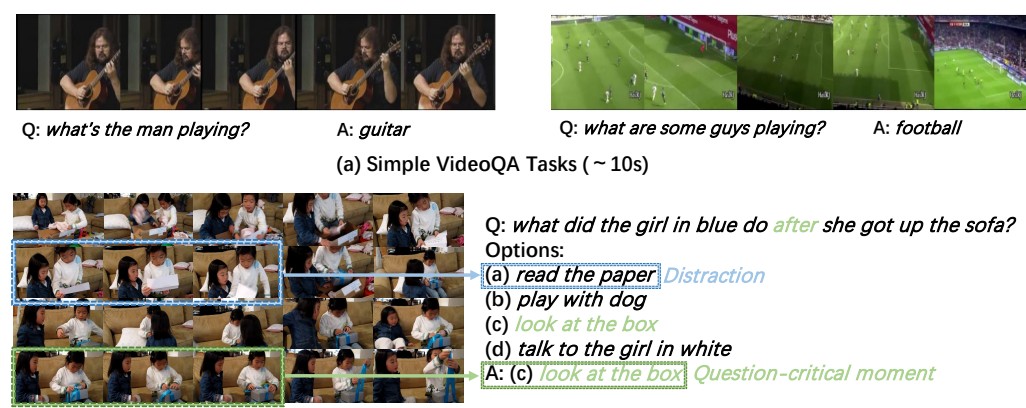

Figure 1: llustration of the distinction between simple VideoQA tasks and complex VideoQA tasks. Simple VideoQA tasks ask descriptive questions such as the location (where is), objects/attributes (who/what (color) is), that can be answered even by one or two frames (like the [guitar] and [football] in (a)), while complex VideoQA tasks aim to explore the logic and commonsense reasoning ability in different scenarios, featuring various causal relationships and temporal dynamics.

integrating recognition-level perception and cognition-level reasoning (Xiao et al., 2021; Li et al., 2022a; Wu et al., 2021). For example, to answer the question in Figure 1 (b), the model should focus on the question-critical moments (bounded with green) corresponding to the phrase [after she got up the sofa] in the question, and then select the correct answer [look at the box], instead of being distracted by the question-irrelevant event [read the paper] in the options, corresponding to a different moment in the video (bounded with blue). As a result, the sparse frames sampling strategy falls short in long videos (30∼60s) containing multiple objects and events, which leads to a deficiency of essential information and can not establish causal-temporal correlations between the key event and general contexts. Although a dense frames sampling strategy (e.g., 32 frames for each video) can capture rich and sufficient information, it introduces its own set of challenges. For one thing, simply concatenating all frames (Figure 2 (b)) without selection results in an unwieldy large number of visual tokens fed into the language model (32×32 visual tokens if concatenating 32 frames, with each frame having 32 tokens), imposing huge computational complexity. For another, the concatenation without selection treats all events in the video equally and leads to the redundancy from large amounts of question-irrelevant frames, distracting the model from discovering the key event required to correctly answer questions.

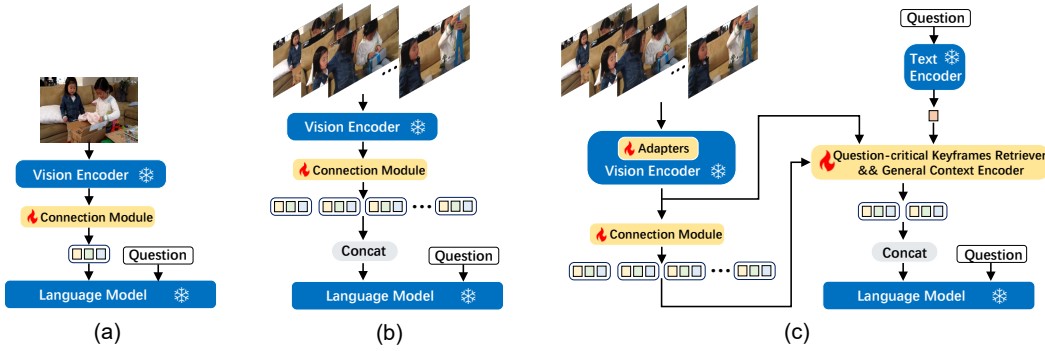

Figure 2: The architecture of MLLM is shown in (a), consisting of a frozen vision encoder, a trainable connection module, and a frozen language model. (b) is the existing method in (Dai et al., 2023; Zhang et al., 2023), simply concatenating uniformly sampled frame representations together. (c) is our method, extending MLLM to be event-aware for spatial-temporal reasoning.

To address these challenges, we propose *E-STR*, extending MLLM to be *E*vent-aware for *S*patial-*T*emporal *R*easoning, as shown in Figure 2 (c). *E-STR* comes from a concise and clear idea, that is, not all events in the video should be equally considered, and the model only needs to retrieve the question-critical event while just viewing other events as general contexts. It works in two folds: Firstly, focusing on question-critical moments can avoid distraction from large amounts of redundant frames. Secondly, the retrieving can effectively reduce the large number of visual tokens caused by the dense frame sampling. Thus, *E-STR* retrieves question-conditioned continuous frames as the key event with a differentiable question-critical keyframes retriever, and compresses unselected frames into a short and coarse-grained representation as general contexts with the general context encoder. Additionally, we insert trainable adapters in the frozen image encoder to better learn the spatial-temporal representations of frames. In summary, we make the following contributions:

1. We explore how to utilize a pre-trained MLLM for complex VideoQA tasks, and we identify the importance and necessity of discovering question-critical moments, which can avoid distraction from redundant contents and reduce the number of visual tokens for efficient computation.

2. We introduce *E-STR*, a novel method to adaptively retrieve question-critical frames and obtain spatial-temporal representations, which is not only straightforward to implement but also cost-effective, enabling MLLM to excel in complex VideoQA tasks.

3. We conduct extensive experiments to verify the effectiveness of our method. Our model outperforms the state-of-the-art results on several complex VideoQA benchmarks including NExT-QA (Xiao et al., 2021), Causal-VidQA (Li et al., 2022a), and STAR (Wu et al., 2021). Further ablation studies verify the effectiveness of each component in our approach.

## 2 RELATED WORK

**Complex video question answering.** VideoQA is the task of answering specific questions related to given videos. Early VideoQA tasks (Jang et al., 2017b; Xu et al., 2017) primarily centered on short video clips about daily lives, with given questions rarely going beyond a recognition of the objects and actions. In terms of this, recently proposed VideoQA benchmarks (Xiao et al., 2021; Li et al., 2022a; Wu et al., 2021) emphasize complex causal and temporal reasoning in long videos, which demand models to acquire a holistic understanding of videos and engage in logical reasoning across intricate real-world scenarios. Recent research has proposed various models to tackle complex VideoQA tasks. Leveraging the capabilities of Transformer (Vaswani et al., 2017) and Graph Neural Networks, some works (Li et al., 2023b; Gao et al., 2023; Xiao et al., 2022b; Jiang & Han, 2020) have designed specific end-to-end models to capture cross-modal motion-appearance interactions. Besides, with the great success of pre-trained techniques, there are also works (Wang et al., 2023b; Bain et al., 2021; Fu et al., 2021) directly fine-tuning pre-trained video-text models on downstream VideoQA tasks. Compared to previous works, this paper is an early exploration of transferring an existing MLLM to complex video-language reasoning tasks.

**Multi-modal large language models.** MLLM (Li et al., 2023a; Alayrac et al., 2022; Dai et al., 2023; Zhu et al., 2023; Ye et al., 2023) are capable of understanding images and language and have shown strong ability in context comprehension and commonsense reasoning. These MLLMs achieve this by adapting frozen language models to frozen image encoders with trainable connection modules as shown in Figure 2 (a). The connection module can either be a simple linear layer (Merullo et al., 2023; Liu et al., 2023) or a transformer-based architecture (Li et al., 2023a; Ye et al., 2023). These MLLMs bridge the modality gap between images and language based on pretraining of the connection module with large-scale image-text data (Gao et al., 2020; Lin et al., 2014; Changpinyo et al., 2021). Our work utilizes MLLM for complex VideoQA tasks, by discovering the key event to avoid distractions and reduce the large number of visual tokens caused by dense frame sampling.

**Transfer learning from image to video.** Efforts in this domain aim to transfer the rich visual knowledge in pre-trained image models to video domains with a small number of trainable parameters, which can be categorized into two kinds. One of a kind is post-temporal modeling, exemplified by previous works such as (Ju et al., 2022; Luo et al., 2022; Bain et al., 2021). They extract each frame's representations independently with the frozen image encoder, then apply techniques like mean pooling, LSTM, and transformers to model the temporal relationships between these individual frame representations to generate a consolidated video representation for downstream video tasks. Another category of works is based on adapters, represented by approaches like ST-Adapter

(Pan et al., 2022), AdapterFormer (Chen et al., 2022), and AIM (Yang et al., 2023). They insert small-sized tunable parameters like MLP or 3D-Convolution kernels into the frozen image encoder, which is expected to learn spatial-temporal representations of video frames. These approaches primarily concentrate on classification tasks like video-text retrieval or video action recognition, while our paper focuses on generative tasks, requiring reasoning ability beyond simple recognition.

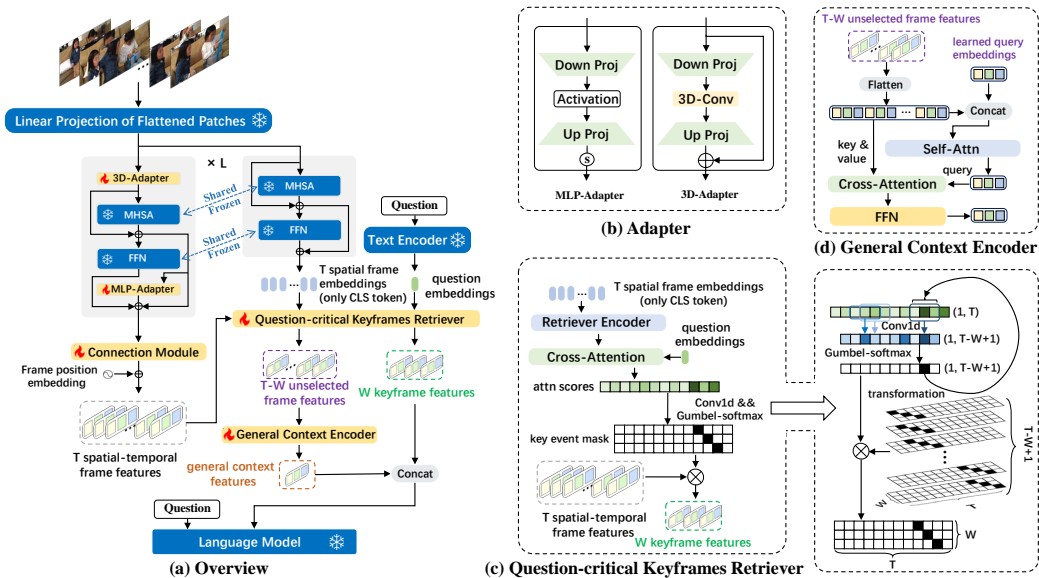

Figure 3: As shown in (a) overview, we extract spatial-temporal frame features $\mathbf{V}_{\mathrm{ST}}$ and spatial frame embeddings $\mathbf{E}_{\mathrm{S}}$ through a dual-path vision encoder. We use the $\mathbf{E}_{\mathrm{S}}$ and question embeddings $\mathbf{E}_{\mathrm{Q}}$ to select the most question-critical part $\mathbf{V}_{\mathrm{Key}}$ within $\mathbf{V}_{\mathrm{ST}}$ with our question-critical keyframes retriever in (c). Besides, we use the general context encoder in (d) to encode the unselected parts $\mathbf{V}_{\mathrm{U}}$ into compressed tokens $\mathbf{V}_{\mathrm{Context}}$ as general contexts. At last, $\mathbf{V}_{\mathrm{Context}}$ and $\mathbf{V}_{\mathrm{Key}}$ will be concatenated to be fed into the language model together with the question to predict the answer.

## 3 METHODOLOGY

In Figure 3 (a), we present the overview of our proposed *E*vent-aware MLLM for *S*patial-*T*emporal *R*easoning, *E-STR*. MLLM with *E-STR* works in four steps: 1) utilize a frozen vision encoder with trainable adapters to extract the spatial-temporal features of the dense frames. 2) identify the question-critical moment to obtain the key frame features by feeding the spatial frame embeddings and the question embeddings into the question-critical keyframes retriever. 3) encode a large number of unselected frame features into coarse-grained short features, serving as general contexts. 4) concatenate the key frame features, general context features, and the question itself, which are then fed into the frozen language model to make the final predictions.

### 3.1 EXTRACT SPATIAL-TEMPORAL FRAME FEATURES

The vanilla vision encoder in MLLM is a standard vision transformer (Dosovitskiy et al., 2021), which can only extract spatial features of each frame independently and is unable to capture temporal relationships between frames. To incorporate temporal modeling, we introduce the 3D-Conv based ST-Adapter (Pan et al., 2022) within each transformer block, which inserts a depth-wise 3D convolution layer between two projections with a residual path, as shown on the right of Figure 3 (b). This ST-Adapter takes tokens for all frames to enable the model to capture temporality. To jointly tune the representations for spatial-temporal reasoning, we also add a lightweight linear-based MLP-Adapter (shown on the left of Figure 3 (b)) at the bottom of each transformer block. $\mathbf{X}_l$, the output of the $l$-th block in vision transformer, can be expressed as:

$$\mathbf{S}_l = \text{ST-Adapter}(\mathbf{X}_{l-1}), \quad \mathbf{H}_l = \mathbf{S}_l + \text{MHSA}(\text{LN}(\mathbf{S}_l)),$$
$$\mathbf{Z}_l = \mathbf{H}_l + \text{FFN}(\text{LN}(\mathbf{H}_l)), \quad \mathbf{X}_l = \mathbf{H}_l + \mathbf{Z}_l + s \cdot \text{MLP-Adapter}(\mathbf{H}_l) \tag{1}$$

where LN, MHSA, FFN respectively mean the layer normalization, multi-heads self-attention, and feed-forward network consisting of two linear layers. $s$ is a scaling factor to control the weight of the output from MLP-Adapter.

We denote the output of the last block as $\mathbf{X}_L \in \mathbb{R}^{T \times N_I \times D_I}$, where L is the number of blocks of the vision encoder, $T$ is the number of frames, $N_I$ is the patch number of each frame (including the class token), and $D_I$ is the embedding dimension. We proceed to feed $\mathbf{X}_L$ into the connection module and add the learned frame positional embeddings on the output of the connection module to obtain the spatial-temporal frame features $\mathbf{V}_{ST} \in \mathbb{R}^{T \times N_C \times D_C}$. $N_C$ is the number of visual tokens of each frame ($N_C \ll N_I$, e.g., $N_C = 32$ and $N_I = 257$ in InstructBLIP (Dai et al., 2023)), and $D_C$ is the hidden size of the connection module. We also obtain the spatial frame embeddings $\mathbf{E}_S \in \mathbb{R}^{T \times D_I}$, extracted from the branch of completely frozen vision encoder and only reserve the class tokens, which will also be used to select the question-critical event. It's notable that the original parameters of the vision encoder are totally frozen and shared, and the only added and trainable parameters come from the lightweight adapters.

### 3.2 QUESTION-CRITICAL KEYFRAMES RETRIEVER

The question-critical keyframes retriever is responsible for selecting $W$ continuous question-critical spatial-temporal frame features $\mathbf{V}_{Key} \in \mathbb{R}^{W \times N_C \times D_C}$ from $\mathbf{V}_{ST} \in \mathbb{R}^{T \times N_C \times D_C}$, as the key event. Here, we emphasize that $W \ll T$, for example, $W = 5$ and $T = 32$.

We firstly feed the spatial frame embeddings $\mathbf{E}_S \in \mathbb{R}^{T \times D_I}$ into the retriever encoder shown in Figure 3 (c), a standard transformer encoder composed of self-attention and FFN. For brevity, we still denote the contextualized outputs of the encoder with $\mathbf{E}_S$. Then, to provide the information of the question, we employ a frozen text encoder to get the question embeddings $\mathbf{E}_Q \in \mathbb{R}^{1 \times D_I}$, of which sequence length is 1 because we only reserve the class token. After that, a cross-attention layer is applied by taking the contextual $\mathbf{E}_S$ as query and the $\mathbf{E}_Q$ as key, which yields the cross-attention map $M \in \mathbb{R}^T$, recording the attention scores between each contextualized frame and the question. To get an event-level attention map, we apply one-dimensional convolution with a window size of $W$ to operate on the $M \in \mathbb{R}^T$, resulting in $\hat{M} \in \mathbb{R}^{T-W+1}$. Each entry $\hat{M}_i$ in $\hat{M}$ is computed as $\hat{M}_i = \sum_{j=i}^{i+W-1} M_j, i \in \{1, \cdots, T-W+1\}$ (each index $i$ in $\hat{M}$ is corresponding to $W$ continuous indices $[i, i+1, \cdots, i+W-1]$ in $M$).

Our goal is to select the index $s$ with the highest score in $\hat{M} \in \mathbb{R}^{T-W+1}$ and subsequently, extract the corresponding $W$ continuous features in $\mathbf{V}_{ST}$, regarded as the most question-critical key event. In detail, We denote this selection as a process that returns the indices of the highest $W$ entries:

$$s = \text{Argmax}(\hat{M}) \in \mathbb{R}^1, \quad y = [s, s+1, \cdots, s+W-1] \in \mathbb{R}^W \tag{2}$$

To perform frame features selection using matrix multiplication, we transform $y$ into a stack of $W$ one-hot $T$-dimensional vectors $Y = [I_{y_1}, I_{y_2}, \cdots, I_{y_W}] \in \{0,1\}^{W \times T}$ as key event mask. This allows us to obtain $\mathbf{V}_{Key} = Y\mathbf{V}_{ST}$. This process is non-differentiable because both $\text{Argmax}$ and one-hot operations are non-differentiable. To learn the parameters of the question-critical keyframes retriever using end-to-end training, we resort to the Gumbel-Softmax (Jang et al., 2017a).

In particular, given the event-level attention map $\hat{M} \in \mathbb{R}^{T-W+1}$, we add the noise $G_i$ sampled from the Gumble distribution to obtain $\widetilde{M} = \left[\hat{M}_1 + G_1, \cdots, \hat{M}_{T-W+1} + G_{T-W+1}\right]$, and replace $\text{Argmax}$ with the differentiable softmax to operate on $\widetilde{M}$:

$$\sigma(\widetilde{M}_i) = \frac{e^{\widetilde{M}_i/\tau}}{\sum_{j=1}^{T-W+1} e^{\widetilde{M}_j/\tau}} \tag{3}$$

Here, $\tau$ represents the temperature parameter, and a smaller value of $\tau$ makes $\widetilde{M}$ closer to a one-hot vector. Having obtained the one-hot $\widetilde{M} \in \mathbb{R}^{T-W+1}$, we multiply it with a constant transformation matrix $H \in \mathbb{R}^{(T-W+1) \times W \times T}$ to get the key event mask $Y = \widetilde{M}H \in \{0,1\}^{W \times T}$. The transforma-

tion matrix is decided by the constant $W$ and $T$ :

$$
\begin{aligned}
H &= [K_1, K_2, \cdots, K_{T-W+1}] \in \mathbb{R}^{(T-W+1) \times W \times T} \\
K_i &= [I_{i,1}, I_{i,2}, \cdots, I_{i,W}] \in \mathbb{R}^{W \times T} \\
I_{i,j} &= [x_1, x_2, \cdots, x_T] \in \mathbb{R}^T, x_t = 1 \text{ if } t = i + j - 1 \text{ else } x_t = 0
\end{aligned}
\tag{4}
$$

This enables the extraction of $\mathbf{V}_{\text{Key}} = Y\mathbf{V}_{\text{ST}}$ in a differentiable manner for end-to-end training.

### 3.3 GENERAL CONTEXT ENCODER

Since the unselected frame features $\mathbf{V}_{\text{U}} \in \mathbb{R}^{(T-W) \times N_{\text{C}} \times D_{\text{C}}} = \mathbf{V}_{\text{ST}} \setminus \mathbf{V}_{\text{Key}}$ provide general contexts of the whole video, we shouldn't discard them. Instead, we use a general context encoder to map it into a much shorter representation $\mathbf{V}_{\text{Context}} \in \mathbb{R}^{N_{\text{C}} \times D_{\text{C}}}$ with only $N_{\text{C}}$ visual tokens. Although the mean pooling of $\mathbf{V}_{\text{U}}$ can do the same thing, we will demonstrate that our general context encoder outperforms it by preserving temporal contexts more effectively in ablation studies.

The general context encoder, depicted in Figure 3 (d), employs a mechanism inspired by the perceiver (Jaegle et al., 2022). It begins by introducing a fixed number of learnable query embeddings $\mathbf{E}_{\text{R}} \in \mathbb{R}^{N_{\text{C}} \times D_{\text{C}}}$, which will be concatenated with the flattened $\mathbf{V}_{\text{U}}$ to be passed through a self-attention layer to obtain a contextualized $\hat{\mathbf{E}}_{\text{R}} \in \mathbb{R}^{N_{\text{C}} \times D_{\text{C}}}$. Next, the contextualized $\hat{\mathbf{E}}_{\text{R}}$ is fed into a cross-attention layer, where it serves as the query while the flattened $\mathbf{V}_{\text{U}}$ is used as both key and value. This operation is followed by a feed-forward network to obtain $\mathbf{V}_{\text{Context}}$:

$$
\hat{\mathbf{E}}_{\text{R}} = \text{Self-Attn}([\mathbf{E}_{\text{R}}; \mathbf{V}_{\text{U}}]), \quad \mathbf{V}_{\text{Context}} = \text{FFN}(\text{Cross-Attn}(Q = \hat{\mathbf{E}}_{\text{R}}, K = V = \mathbf{V}_{\text{U}}))
\tag{5}
$$

### 3.4 SPATIAL-TEMPORAL REASONING

At last, the obtained $\mathbf{V}_{\text{Context}} \in \mathbb{R}^{N_{\text{C}} \times D_{\text{C}}}$ and $\mathbf{V}_{\text{Key}} \in \mathbb{R}^{W \times N_{\text{C}} \times D_{\text{C}}}$ are concatenated, to get a video representation with $W + 1$ tokens, together with the question $\mathcal{Q}$ to be fed into the frozen language model for final reasoning. The model is trained using the cross-entropy loss with parameters $\theta$:

$$
\mathcal{L} = -\sum_{t=1}^{L_a} log P_\theta(\mathcal{A}_t | \mathcal{A}_{<t}, \mathbf{V}_{\text{Context}}, \mathbf{V}_{\text{Key}}, \mathcal{Q})
\tag{6}
$$

where $\mathcal{A}_t$ is predicted autoregressively at position $t$, and $L_a$ is the sequence length of the ground truth answer text $\mathcal{A}$.

## 4 EXPERIMENTS

### 4.1 EXPERIMENTAL SETUP

**Datasets.** We evaluate our *E-STR* on NExT-QA (Xiao et al., 2021), Causal-VidQA (Li et al., 2022a), and STAR (Wu et al., 2021). Specifically, NExT-QA contains 5.4k videos with an average length of 44s and proposes 52k questions, including description, explanation, and temporal reasoning questions. The Causal-VidQA selects 27k video clips and asks 108k questions, including description, explanation, prediction, and counterfactual questions. STAR is proposed for situated reasoning, containing 22K video clips along with 60K questions. All of them use a multi-choice setting, which aims to test the temporal reasoning ability with complex causal and commonsense relations. For each benchmark, the standard answer accuracy is adopted as the metric.

**Baselines.** In this paper, we choose the InstructBLIP (Dai et al., 2023) as our MLLM, which adopts the ViT-G in EVA-CLIP (Fang et al., 2023) as the image encoder, a transformer-based Q-former (Li et al., 2023a) as the connection module, and the Vicuna (Zheng et al., 2023) or FLAN-T5 (Chung et al., 2022) as the language model. In addition to previous works proposed for complex VideoQA tasks, we also use the InstructBLIP with concatenation and mean-pooling of the uniformly sampled frame features as in (Dai et al., 2023; Luo et al., 2022) as our baselines for a fair comparison.

**Implementation details.** We uniformly sample $T = 32$ frames per video, and each frame is cropped into a size of 224×224. The length of visual tokens is set to $N_C = 32$. Besides, we set the event window size $W = 5$. For the text encoder, we adopt the pre-trained text encoder in the same

| Method | NExT-QA | | | | Causal-VidQA | | | | |
|---|---|---|---|---|---|---|---|---|---|
| | @Des | @Tem | @Cau | @All | @Des | @Exp | @Pre | @Cou | @All |
| Co-Mem (Gao et al., 2018) | 54.4 | 50.0 | 45.9 | 48.5 | 64.1 | 62.8 | 31.4 | 32.6 | 47.7 |
| HCRN (Le et al., 2020) | 54.0 | 49.3 | 47.1 | 48.9 | 56.4 | 61.6 | 32.6 | 32.7 | 48.1 |
| HGA (Jiang & Han, 2020) | 57.8 | 49.1 | 48.1 | 50.0 | 65.7 | 63.5 | 32.2 | 34.3 | 48.9 |
| IGV (Li et al., 2022b) | 59.6 | 51.7 | 48.6 | 51.3 | 65.9 | 62.1 | 35.0 | 31.2 | 48.6 |
| HQGA (Xiao et al., 2022a) | 59.4 | 52.3 | 49.0 | 51.8 | - | - | - | - | - |
| B2A (Park et al., 2021) | 58.3 | 49.0 | 47.4 | 49.6 | 66.2 | 62.9 | 31.2 | 35.2 | 49.1 |
| MCR (Zang et al., 2023) | 62.3 | 52.0 | 49.2 | 52.4 | 67.5 | 65.6 | 37.8 | 33.4 | 51.1 |
| MIST (Gao et al., 2023) | 66.9 | 56.6 | 54.6 | 57.2 | - | - | - | - | - |
| TranSTR (Li et al., 2023b) | 70.0 | 60.2 | 59.7 | 61.5 | 73.6 | _75.8_ | 48.9 | _50.3_ | _62.2_ |
| VQA-T* (Yang et al., 2021) | 63.2 | 51.5 | 49.6 | 52.3 | - | - | - | - | - |
| VGT-PT* (Xiao et al., 2022b) | 67.3 | 54.5 | 52.8 | 55.7 | 70.8 | 70.3 | 38.4 | 42.0 | 55.4 |
| HiTeA* (Ye et al., 2022) | 75.6 | 62.4 | 58.3 | 63.1 | - | - | - | - | - |
| InternVideo* (Wang et al., 2022) | 75.8 | 62.5 | 58.5 | 63.2 | - | - | - | - | - |
| InstructBLIP (Vicuna-13B) | _78.9_ | _67.2_ | _67.9_ | _69.5_ | **78.6** | 73.6 | _52.2_ | 43.6 | 61.9 |
| + E-STR | **80.2** | **69.3** | **72.6** | **72.8** | _77.6_ | **77.4** | **57.2** | **51.1** | **65.8** |

Table 1: Accuracy (%) on NExT-QA and Causal-VidQA. @Des, @Tem, and @Cau denote questions type of Descriptive, Temporal, and Causal in NExT-QA. @Des, @Exp, @Pre and @Cou denote questions type of Description, Explanation, Prediction, and Counterfactual in Causal-VidQA. * means methods with large-scale video-text pertaining. The **best** and _2nd-best_ results are highlighted.

| Method | STAR | | | | |
|---|---|---|---|---|---|
| | @Int | @Seq | @Pre | @Fea | @All |
| ClipBERT (Lei et al., 2021) | 39.8 | 43.6 | 32.3 | 31.4 | 36.7 |
| CLIP (1 frame) (Radford et al., 2021) | 39.8 | 40.5 | 35.5 | 36.0 | 38.0 |
| Flamingo-9B (Alayrac et al., 2022) | - | - | - | - | 43.4 |
| AIO (Wang et al., 2023a) | 47.5 | 50.8 | 47.8 | 44.1 | 47.5 |
| MIST (Gao et al., 2023) | 55.6 | 54.2 | 54.2 | 44.5 | 51.1 |
| InternVideo (Wang et al., 2022) | 62.7 | 65.6 | 54.9 | _51.9_ | 58.7 |
| InstructBLIP (Vicuna-13B) | _65.4_ | _68.6_ | _59.1_ | 51.8 | _61.2_ |
| + E-STR | **69.1** | **72.0** | **61.2** | **56.3** | **64.7** |

Table 2: Accuracy (%) on STAR. @Int, @Seq, @Pre, and @Fea denote questions type of Interaction, Sequence, Prediction, and Feasibility in STAR, respectively. The **best** and _2nd-best_ results are highlighted.

| Models | NExT-QA | | | |
|---|---|---|---|---|
| | @Des | @Tem | @Cau | @All |
| FLAN-T5-XL◇ | 78.5 | 65.6 | 69.2 | 69.6 |
| FLAN-T5-XXL◇ | 80.7 | 66.3 | 70.2 | 70.7 |
| Vicuna-7B◇ | 77.2 | 65.5 | 67.3 | 68.4 |
| Vicuna-13B◇ | 78.9 | 67.2 | 67.9 | 69.5 |
| FLAN-T5-XL♦ | _80.6_ | 66.3 | 70.2 | 71.4 (+1.8) |
| FLAN-T5-XXL♦ | 79.1 | **71.8** | **74.8** | **74.5** (+3.8) |
| Vicuna-7B♦ | 79.0 | 67.9 | 72.1 | 71.7 (+3.3) |
| Vicuna-13B♦ | 80.2 | _69.3_ | _72.6_ | _72.8_ (+3.3) |

Table 3: Accuracy (%) on NExt-QA of InstructBLIP with different language models. ◇ means vanilla models with concatenation of 6 uniformly sampled frames. ♦ means models with E-STR.

EVA-CLIP. During training, we keep the parameters of the image encoder, language model, and text encoder frozen. We use AdamW to optimize the model with a learning rate of $2e^{-5}$ and the strategy of mixed precision. We run our experiments on 4 NVIDIA A100 GPUs.

## 4.2 MAIN RESULTS

**Comparison with State-of-the-arts.** In Table 1 and 2, results show that we outperform current methods on all three benchmarks by a significant margin (NExT-QA+3.3%, Causal-VidQA+3.6%, and STAR+3.5%). Notably, we observe that the improvement is largely from complex questions (@Tem and @Cau in NExT-QA, @Exp and @Pre and @Cou in Causal-VidQA) that require an understanding of causal relations and temporal reasoning. This demonstrates the outstanding spatial-temporal reasoning ability of our method for complex VideoQA tasks. We also conduct experiments to compare different MLLMs with various language models. As shown in Table 3, we still achieve superior performance on NExT-QA and gain a large improvement over vanilla models with concatenation of 6 uniformly sampled frames, demonstrating the effectiveness and generalizability of our method across different vision-language models. In the following experiments, we use InstructBLIP-Vicuna-7B as our default model and NExT-QA as the default benchmark unless otherwise specified.

**Comparison with baselines.** To ensure a fair comparison, we also select vanilla models with both concatenation and mean-pooling as our baselines. Table 4 reveals that the performance of mean-pooling is considerably lower than that of concatenation with the same number of frames (66.7% vs. 71.1% with 32 frames). Furthermore, as the frame count increases, the performance gains in both concatenation and mean pooling diminish (+ 0.2% from Concat-24 to Concat-32). Notably, our proposed method consistently outperforms both baselines significantly (71.7% vs. 71.1% vs. 66.7%), demonstrating the necessity of discovering question-critical moments.

| Method | Frame Num. | Visual Tokens Length | Tunable Param.(M) | GFLOPs | NExT-QA @Des | @Tem | @Cau | @All |
|---|---|---|---|---|---|---|---|---|
| Concat-6 | 6 | 6×32 | 188 | 9382 | 77.2 | 65.5 | 67.3 | 68.4 |
| Concat-12 | 12 | 12×32 | 188 | 10330 | 77.9 | 66.6 | 68.7 | 69.6 |
| Concat-24 | 24 | 24×32 | 188 | 12094 | 78.6 | 67.3 | 70.5 | 70.9 |
| Concat-32 | 32 | 32×32 | 188 | 15616 | 78.0 | 67.6 | 71.1 | 71.1 |
| Mean-6 | 6 | 1×32 | 188 | 8610 | 75.2 | 60.3 | 61.5 | 63.4 |
| Mean-12 | 12 | 1×32 | 188 | 8712 | 76.4 | 62.8 | 63.9 | 65.6 |
| Mean-24 | 24 | 1×32 | 188 | 8882 | 76.5 | 63.8 | 65.1 | 66.6 |
| Mean-32 | 32 | 1×32 | 188 | 9116 | 76.3 | 64.1 | 65.2 | 66.7 |
| Ours | 32 | (5+1)×32 | 237 | 9673 | 79.0 | 67.9 | 72.1 | **71.7** |

Table 4: Comparison with the performance and computation efficiency of baselines on NExt-QA. The GFLOPs are tested on a single GPU with a batch size = 1 during inference for one step.

**Computation efficiency.** In Table 4, it's evident that accuracy increases when more frames are sampled with the concatenation method. However, this improvement comes at the cost of more visual tokens (32×32) and greater computational complexity (15616 GFLOPs). In contrast, mean-pooling, while less computationally demanding (9116 GFLOPs), yields lower performance (66.7%). Our approach strikes a balance between these factors, achieving better performance than Concat-32 (71.7% vs. 71.1%) while incurring close computational costs as Concat-6 (9673 GFLOPs vs. 9382 GFLOPs). We attribute it to the smaller number of visual tokens (6×32), significantly reducing the computational burden from the large language models.

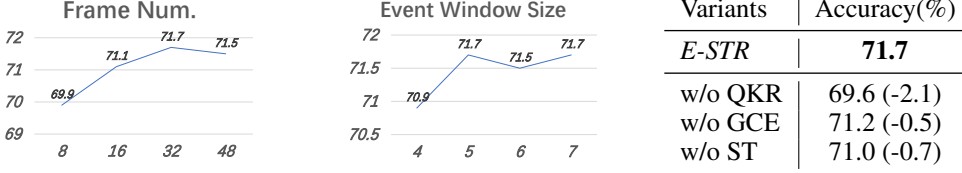

| Variants | Accuracy(%) |
|---|---|
| *E-STR* | **71.7** |
| w/o QKR | 69.6 (-2.1) |
| w/o GCE | 71.2 (-0.5) |
| w/o ST | 71.0 (-0.7) |

Figure 4: Ablation study of different numbers of frames.

Figure 5: Ablation study of different event window sizes.

Table 5: Ablation study of each component.

## 4.3 ABLATION STUDIES

**Number of frames.** We evaluate how the number of sampled frames $T$ will influence the performance with a fixed event window size $W = 5$. The results shown in Figure 4 indicate that performance improves as more frames are included, up to a point $T = 32$. However, beyond a certain threshold ($T = 48$), there is a performance drop. This suggests that too many frames may introduce redundancy and noise, while too few frames miss important information and interrupt the continuity of events, which are critical to the question.

**Event window size.** Both the computation efficiency and the number of frames directly fed into the language model are dependent on the event window size $W$. As depicted in Figure 5, when the number of frames is fixed to $T = 32$, a smaller $W$ captures finer-grained temporal information but may miss some relevant frames, while a larger $W$ provides a broader context but introduces more noise. Our model achieves the best performance when $W = 5$, demonstrating that 5 continuous frames can represent a key event most appropriately.

**Effect of each component.** To assess the individual contributions of each component, we perform ablation studies by removing specific modules one at a time. In Table 5, **w/o QKR** means we replace the Question-critical Keyframes Retriever (QKR) with uniformly sampled $W$ frames, and **w/o GCE** means we replace the General Context Encoder (GCE) with mean-pooling. In **w/o ST**, we remove the adapters in the image encoder to discard the Spatial-Temporal (ST) frame features and only use spatial frame features for both retrieval and reasoning. In our results, we find that all of them contribute positively to the model's performance. Among them, the question-critical keyframes

retriever has the most significant impact (-2.1%), underlining its crucial role in discovering question-critical moments for the avoidance of distractions in reasoning.

**The choice of text encoder.** We employ various pre-trained language models to encode the questions, which include EVA-CLIP (Fang et al., 2023), CLIP (Radford et al., 2021), BERT (Devlin et al., 2019), and the encoder of FLAN-T5 (Chung et al., 2022). Notably, the results in Table 6 demonstrate that the text encoder of EVA-CLIP yields the highest performance. This is intuitively understandable since the question embeddings $\mathbf{E}_Q$, derived from the text encoder of EVA-CLIP, naturally align well with the spatial frame embeddings $\mathbf{E}_S$, extracted with the image encoder in the same EVA-CLIP. This alignment is due to the fact that both the text encoder and the image encoder in EVA-CLIP have undergone large-scale contrastive learning, which encourages them

| Text Enc. | Accuracy(%) |
|---|---|
| EVA-CLIP | **71.7** |
| CLIP | 70.9 |
| BERT | 70.6 |
| FLAN-T5 | 70.5 |

Table 6: Ablation study of different text encoder.

to have a shared understanding of the visual and textual content. This shared understanding likely contributes to the superior performance observed when using the EVA-CLIP text encoder.

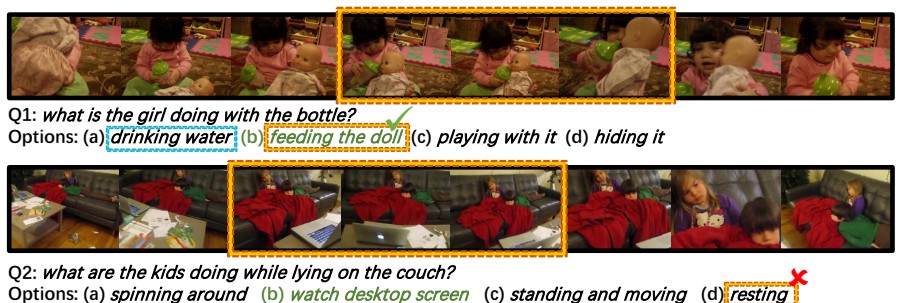

Q1: *what is the girl doing with the bottle?*
Options: (a) *drinking water* (b) *feeding the doll* (c) *playing with it* (d) *hiding it*

Q2: *what are the kids doing while lying on the couch?*
Options: (a) *spinning around* (b) *watch desktop screen* (c) *standing and moving* (d) *resting*

Figure 6: Qualitative results on NExT-QA test set, the frames of key events selected by our question-critical keyframes retriever are highlighted in orange, together with our predictions. The option highlighted in blue is the wrong prediction of the model with a concatenation strategy. The ground truth is in green. **Q1** presents the successful case, while **Q2** is a failure case.

## 4.4 QUALITATIVE RESULTS

In Figure 6, we present qualitative results to gain a clearer insight into our method. The upper case demonstrates scenarios where the correct answer is only possible with our method. For instance, in **Q1**, our question-critical keyframes retriever correctly captures the critical continuous frames, describing the event of [`feeding the doll`]. Instead, the model chooses the option of [`drinking water`] incorrectly with a concatenation of uniformly sampled frames. However, the bottom case in **Q2** highlights situations where we encounter challenges. Although we locate the moment corresponding to [`watch desktop screen`], we choose the wrong answer [`resting`]. We attribute it to the language bias (Dancette et al., 2021) in the frozen language model, which tends to incorrectly associate the phrase [`lying on the couch`] in the question with the word [`resting`] due to their frequent co-occurrence in the pre-training corpus. These biases are expected to be solved with a better pre-trained MLLM and our method is easy to build upon.

## 5 CONCLUSION

This paper introduces *E-STR*, extending MLLM to be *E*vent-aware for *S*patial-*T*emporal *R*easoning in complex VideoQA tasks. *E-STR* effectively avoids distractions from the question-irrelevant contents and reduces the number of visual tokens resulting from dense frame sampling, by retrieving question-critical frames as the key event and compressing redundant ones as the general contexts. Besides, *E-STR* get spatial-temporal representations of frames with a few lightweight adapters. Through a series of experiments, we have demonstrated the superiority of our method. In the future, we plan to mitigate the potential presence of biases in the data as observed in previous VQA studies, to further enhance the reasoning ability of current models and build a more robust reasoning system.

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

## A  MORE IMPLEMENT DETAILS

We use the PyTorch framework for distributed training, specifically with the DistributedDataParallel module. Batch sizes varied depending on the number of sampled frames: 4 for 32 frames, 8 for 16 frames, 16 for 8 frames, and 32 for 4 frames. The learning rate was set to 2e-5, and the training ran for 10 epochs. For models using Vicuna as the language model, we employ mixed precision training with Float16. For models with FLAN-T5, we use BFloat16. In addition to the connection module (188M), additional trainable parameters in our method stem from the adapters (28M), question-critical keyframes retriever (2M), and general context encoder (19M).

The ViT-G from EVA-CLIP is adopted as the image encoder, removing the last layer and using the second-to-last layer's output features for reasoning. To align the frozen text embeddings with frozen frame embeddings which had been well learned during contrastive pre-training, we add the original last ViT block in the spatial branch. In the MLP-Adapters, we set the down projections to have a shape of (1408, 128) and the up projections with a shape of (128, 1408), with a scaling factor of 0.1, and the activation function is GELU. The 3D-Conv kernel in the 3D-Adapter had input channels set to 1408, output channels to 128, and a kernel size of (3,1,1).

We reformate the dataset prompts as text with a specific structure, including the question and answer options. The prompt structure was designed as follows: "$Question :< Question > Options : (-) < option_0 > (-) < option_1 > (-) < option_2 > (-) < option_3 > Answer :$", where $< Question >$ and $< option_i >$ are all from raw data. We feed this prompt to the frozen language model in MLLM to predict the correct option $< option_j >$ autoregressively.

## B  MORE ABLATION STUDIES

We present additional experiments aimed at further exploring the effectiveness of our method. Specifically, we continue to employ the Vicuna-7B model and the NExT-QA benchmark.

### B.1  THE CONTRIBUTION OF EACH COMPONENT

We conduct comprehensive tests to assess the individual contributions of each component within our approach. The results, displayed in Table 7, highlight the significant improvements achieved through our question-critical keyframes retriever, which selectively extracts 6 continuous question-critical

| QKR | GCE | ST | Accuracy(%) |
|-----|-----|-----|-----|
| ✗ | ✗ | ✗ | 68.4 |
| ✓ | ✗ | ✗ | 70.5 |
| ✗ | ✓ | ✗ | 68.7 |
| ✗ | ✗ | ✓ | 69.3 |
| ✗ | ✓ | ✓ | 69.6 |
| ✓ | ✗ | ✓ | 71.2 |
| ✓ | ✓ | ✗ | 71.0 |
| ✓ | ✓ | ✓ | **71.7** |

Table 7: More detailed ablation studies on each component.

| MLP-Adapter | 3D-Adapter | Accuracy(%) |
|-----|-----|-----|
| ✗ | ✗ | 71.0 |
| ✓ | ✗ | 70.6 |
| ✗ | ✓ | 71.4 |
| ✓ | ✓ | **71.7** |

Table 8: Ablation studies of adapters.

| Hidden Size | Accuracy(%) |
|-----|-----|
| 1024 | 71.4 |
| 768 | 71.5 |
| 512 | **71.7** |
| 256 | 71.2 |

Table 9: Ablation studies of hidden size

| Dataset | QA pair | Video-Lengh |
|-----|-----|-----|
| MSVD-QA | 50k | 10s |
| MSRVTT-QA | 224k | 15s |
| ActivityNet-QA | 58k | 180s |

Table 10: Dataset statistics of simple VideoQA tasks.

frames. InstructBLIP with a single retriever outperforms uniform frame concatenation greatly, elevating accuracy from 68.4% to 70.5%. While a single general context encoder brings modest improvements, it still surpasses mean pooling (from 68.4% to 68.7%), which can result in the loss of valuable temporal information. Furthermore, spatial-temporal representations are also instrumental, both in isolation (from 68.4% to 69.3%) and when combined with other components.

### B.2 SETTINGS OF ADAPTERS

We investigate the optimal configurations of our adapters in this set of experiments. As detailed in Table 8, we observe an increase in accuracy with the introduction of a single 3D-Adapter and further enhancements when combining the 3D-Adapter with the MLP-Adapter. However, the incorporation of a single MLP-Adapter leads to a decline in performance. This outcome can be attributed to the 3D-Adapter's ability to capture temporal information, while the MLP-Adapter's role is to refine the frozen parameters in the MHSA and FFN layers to align better with the 3D-Adapter. In contrast, the single MLP-Adapter struggles to handle temporal relationships between tokens and can interfere with well-established spatial representations.

### B.3 HYPER PARAMETERS OF QUESTION-CRITICAL KEYFRAMES RETRIEVER

To investigate the impact of different hidden sizes of the encoder and cross-attention layer in our question-critical keyframes retriever, we experiment with values of 1024, 768, 512, and 256. The results, presented in Table 9, demonstrate that a hidden size of 512 yields the best performance for our question-critical keyframes retriever.

## C SUPPLEMENTARY EXPERIMENTS ON SIMPLE VIDEOQA TASKS

While not the primary focus of this paper, we applied our method to simple VideoQA tasks, including MSVD-QA (Xu et al., 2017), MSRVTT-QA (Xu et al., 2017), and ActivityNet-QA (Yu et al., 2019), to assess the generalization of our approach. Dataset statistics are provided in Table 10. These tasks utilize an Open-Ended setting and emphasize describing video objects, activities, and their attributes. The answers to these questions are typically short and straightforward, often involving binary responses (e.g., yes/no) or identifying basic attributes (e.g., man/woman). We adopt the same training setting as in complex VideoQA tasks.

| Method | MSVD-QA | MSRVTT-QA | ActivityNet-QA |
|---|---|---|---|
| Co-Mem (Gao et al., 2018) | 34.6 | 35.3 | - |
| HCRN (Le et al., 2020) | 36.1 | 35.6 | - |
| HGA (Jiang & Han, 2020) | 34.7 | 35.5 | - |
| IGV (Li et al., 2022b) | 40.8 | 38.3 | - |
| HQGA (Xiao et al., 2022a) | 41.2 | 38.6 | - |
| VGT (Xiao et al., 2022b) | - | 39.7 | - |
| VQA-T* (Yang et al., 2021) | 46.3 | 41.5 | 38.9 |
| FrozenBiLM* (Yang et al., 2022) | 55.5 | 47.0 | 43.2 |
| HiTea* (Ye et al., 2022) | 55.6 | 45.9 | - |
| InterVideo* (Wang et al., 2022) | 55.5 | 47.1 | - |
| VideoCoCa* (Yan et al., 2022) | 56.9 | 46.3 | **56.1** |
| mPLUG2* (Xu et al., 2023) | 58.1 | 48.0 | - |
| VAST* (Chen et al., 2023b) | 60.0 | - | 50.0 |
| VALOR* (Chen et al., 2023a) | 60.0 | - | 48.6 |
| MaMMUT* (Kuo et al., 2023) | 60.2 | 49.5 | - |
| VLAB* (He et al., 2023) | 61.0 | **49.6** | - |
| InstructBLIP (Vicuna-13B) | 59.6 | 46.9 | 45.9 |
| + *E-STR* | **61.1** | 49.1 | 49.2 |

Table 11: Accuracies (%) on simple VideoQA tasks. * means methods with large-scale video-text pertaining. The **best** and 2nd-best results are highlighted.

Our results are summarized in Table 11. In contrast to complex VideoQA tasks, we observe that methods incorporating video-text pretraining perform significantly better in simple VideoQA tasks. This phenomenon suggests that video-text pretraining can substantially enhance the ability to recognize objects and attributes in videos, a primary focus of simple VideoQA tasks. However, the challenging skill of causal temporal reasoning, essential in complex VideoQA tasks, remains difficult to achieve within the current paradigm of video-text pretraining. Notably, our method still achieves competitive results on these three benchmarks and demonstrates notable improvements over vanilla baseline models. Moreover, it's worth mentioning that our method exhibits more substantial performance gains over vanilla models with uniform concatenation in complex VideoQA tasks (NExT-QA+3.3%, Causal-VidQA+3.9%, and STAR+3.5%), compared to simple VideoQA tasks (MSVD-QA +1.5%, MSRVTT-QA +2.2%, ActivityNet-QA +3.3%). This discrepancy can be attributed to the differences between these task categories. Open-ended datasets (MSVD-QA, MSRVTT-QA, ActivityNet-QA) primarily feature straightforward questions and short videos, whereas complex VideoQA datasets (NExT-QA, Causal-Vid, STAR) focus on intricate questions, long videos, and multifaceted events, necessitating advanced spatial-temporal reasoning capabilities. This precisely reflects the strengths of our method, which identifies question-critical video segments and eliminates redundancy to enhance reasoning.

## D MORE VISUALIZATIONS

In Figure 7, we present additional visualizations of our method's successful prediction results. These cases vividly illustrate the advantages of our approach. For instance, in **Q4**, *E-STR* adeptly captures the sequence of events where the man is smiling while carrying the baby, leading to the correct answer [happy]. Furthermore, leveraging common sense knowledge inherent in the vanilla MLLM, *E-STR* can effectively deduce that the force of [gravity] is responsible for the shirt coming off in **Q5**, all after precisely locating the moment when the event [did a flip] occurs.

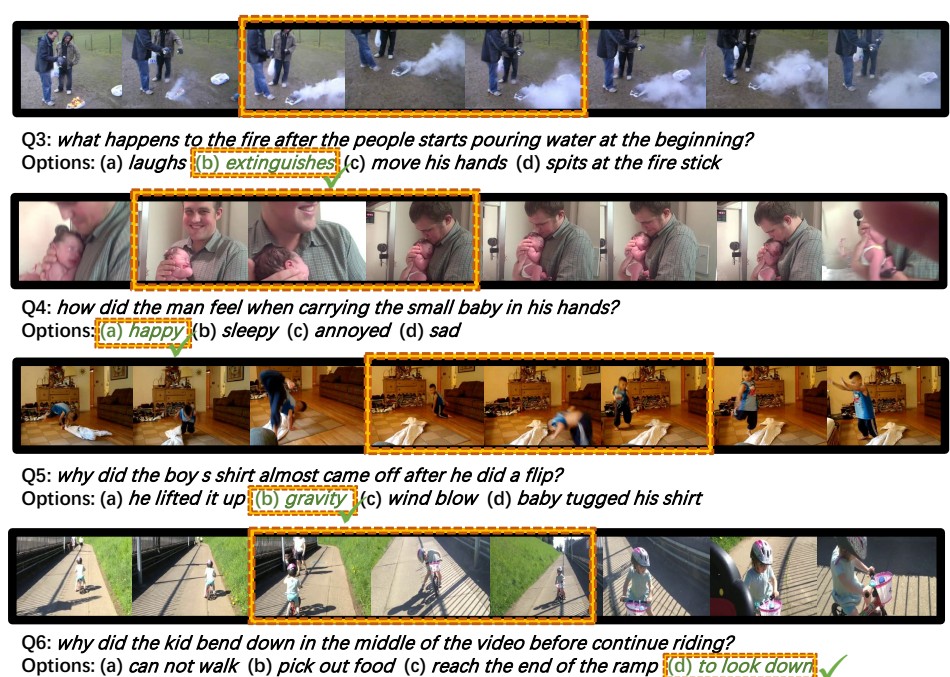

Figure 7: More successful qualitative results on NExT-QA test set.

Figure 8 provides additional examples of cases where our model faces challenges. For instance, the model struggles when questions involve counting entities in videos, as seen in **Q7**. This challenge is particularly pronounced when the entities appear at different moments within the video. Another observed challenge arises when the concept mentioned in the question occurs multiple times in the video. In **Q8**, where there are multiple instances of [black dogs] in the video, the keyframes retriever locates moments when two black dogs are playing together, introducing ambiguity to the model's understanding. We believe that these issues can be alleviated by proposing better pre-trained MLLM, and our method is easy to build upon the stronger ones.

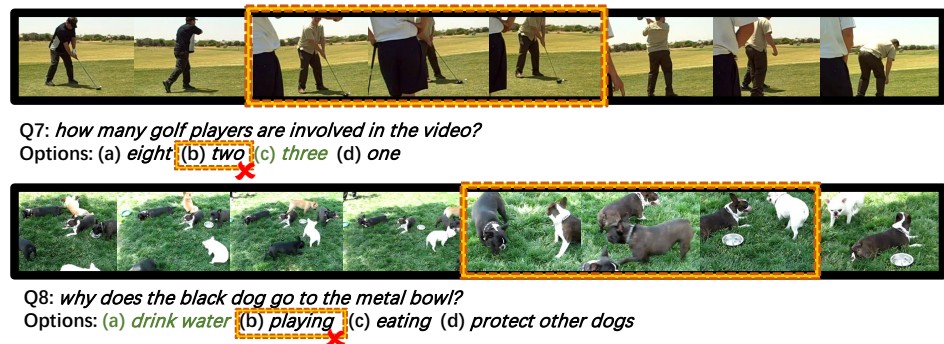

Figure 8: More failure qualitative results on NExT-QA test set.

# E  PSEUDO CODE

As explained in the paper, our method is effective and simple to implement. We show the PyTorch-style pseudo-code on how to implement the question-critical keyframes retriever as follows:

```python
class Question_critical_Keyframes_Retriever ():
    def __init__(self, T, W):
        self.encoder = TransformerEncoder()
        self.cross_attn = CrossAttention()
        self.W = W # event window size
        self.T = T # frame number

    def make_one_hot(self, T, W, index):
        K = torch.zeros(W, T)
        for i in range(W):
            K[i][index+i] = 1
        return K

    def make_transformation_matrix(self, T, W):
        H = []
        for i in range(T-W+1):
            H.append(self.make_one_hot(T, W, i))
        H = torch.stack(H, dim=0)
        return H

    def forward(self, Q, K, V):
        '''
        Q: (bs, 1, D_I) question_embeddings
        K: (bs, T, D_I) spatial frame embeddings
        V: (bs, T, N_C, D_C) spatial-temporal frame features
        '''

        T = self.T
        w = Self.W

        # get contextualized spatial frame embeddings
        K = self.encoder(K) # [bs, T, D_I]

        # get frame-level attn_map
        frame_attn_map = self.cross_attn(query=Q, key=K) # [bs, T]

        # get event-level attn_map with 1D-Conv
        event_attn_map = Conv1d(frame_attn_map, weight=torch.ones(1, 1,
            W), stride=1) # [bs, T-W+1]

        # differential selection with Gumbel_Softmax
        one_hot_mask = Gumbel_softmax(event_attn_map, tau=1, dim=-1,
            hard=True) # [bs, T-W+1]

        # get transformation matrix
        H = self.make_transformation_matrix(T, W) # [T-W+1, W, T]

        # calculate key event mask
        key_event_mask = torch.einsum("b r, r k t -> b k t", one_hot_mask,
            H) # [bs, W, T]

        # select the most critical event from V
        selected_event = torch.einsum("b t n d, b k t -> b k n d", V,
            key_event_mask)
        # [bs, W, N_C, D_C]

        return selected_event
```

