# OpenReview forum: "Discovering the question-critical moments: Towards building event-aware multi-modal large language models for complex video question answering"
_ICLR.cc/2024/Conference — Submitted to ICLR 2024_

### Official Review · Reviewer_zSXn · 2023-10-27

**Soundness:** 2 fair
**Presentation:** 2 fair
**Contribution:** 2 fair
**Rating:** 3
**Confidence:** 4

**Summary:**

The author focused on better-resolving video question answering. Among various approaches, the authors focus primarily on long videos. Although the topic and the focus sound interesting, it is hard to understand the main difference compared to the previous works and how each model is constructed in what manner.

**Strengths:**

First, the author well-presented the problem, and I do not doubt this part. Also, the experiment seems very well designed with multiple datasets. From my end, the justifications in Figure 4 help me understand why the author set each parameter. Last but not least, I did not find any grammatical issues.

**Weaknesses:**

It is very hard to catch what are the main differences between previous approaches. Also, it is almost impossible to understand how each module introduced in Section 3 is constructed, even with Figure 3. As a result, it is tough to find a clear link before Section 4 and after Section 4.

**Questions:**

A. First and foremost, the authors sell their approach as somewhat new in handling long-term video. For instance, the first sentence in 3rd paragraph of the Intro (there are a couple more, e.g., the last paragraph of the Intro and the first paragraph of Related work) treats pioneer video QA tasks/methods do not focus more on causal and temporal inference. However, even from the beginning of VideoQA (Jang et al., 2017b) (I would like to cite some additional work in this thread [1,2,3]), they tackle causality with spatial/temporal attention mechanism (for instance, the oldest baseline Co-Mem (Gao et al., 2018) also uses attention). Considering this work is also mainly based on attention mechanisms, I missed the main difference between these lines of work. The author may want to say that those works are not based on the Transformer model, and it should be true for some old approaches, as those works appeared even before the Transformer was presented, but it is only valid for some of the baselines. Instead of mainly focusing on presenting numbers, I would request to present a detailed analysis with a theoretical explanation, and I do believe this will strengthen this manuscript. Along with this, I also wonder how and why the authors think some Transformer-based approaches that sample a few frames from the vision side (e.g., MERLOT [4,5]) work reasonably well on VideoQA, even though some of those models do not have an explicit attention-based frame sampler. Comparison with those models would also be appropriate.

B. Along with A, it is almost impossible to understand how each component presented in Section 3 is constructed. I guess X_{L} comes from X_{l} in Equation (1), but I failed to find any clue for V_{ST}, V_{Key}, E_{S}, E_{Q}. The only equation I can see afterward is an Argmax in Eq.(2); it is impossible to guess how to compute those. I also failed to see any symbols from tiny Figure 3 (a) (The author should write the main paper self-contained within the page limit). I don't think any reader can easily replicate this work without such details.

Due to A and B, I feel Sections 1-3 and 4-5 are disconnected, and thus, it is hard to fully digest the experiment results; it seems the experiment itself is reasonably designed, by the way. To this end, it is hard to give acceptance from my end as of now. I suggest the authors (aggressively) revise Sections 1-3 to sound more coherent with Section 4~5.


*** References ***

[1] Zhu et al., Uncovering the Temporal Context for Video Question Answering, IJCV 2017.

[2] Mun et al., MarioQA: Answering Questions by Watching Gameplay Videos, in ICCV 2017.

[3] Kim et al., DeepStory: Video Story QA by Deep Embedded Memory Networks, in IJCAI 2017.

[4] Zellers et al., MERLOT: Multimodal Neural Script Knowledge Models, in NeurIPS 2021.

[5] Zellers et al., MERLOT Reserve: Neural Script Knowledge through Vision and Language and Sound, in CVPR 2022.

---

> ### Author Response · Authors · 2023-11-12
>
> We thank the reviewer for the valuable feedback on our paper. In the next paragraphs, we address the comments provided by the reviewer.
>
> **Q1: Differences from previous approaches.**
>
> **(1) We explore how to utilize a pre-trained MLLM for complex VideoQA tasks, instead of designing a new network from scratch.**
>
> As you mentioned in your review, there are many previous works [1,2,3,4] that tackle VideoQA with an attention mechanism. However, works in this line typically **devise independent neural network models from scratch** (e.g., similar to the relationships between Alexnet, Vggnet, and Resnet). Instead, our work is on the basis of a pre-trained MLLM, and our motivation is to enhance its video-language reasoning ability through our method.
>
> **(2) We utilize the attention mechanism for the hard selection of frame features, instead of the weighted summation of frame features.**
>
> As we concluded **in the 2nd paragraph, Sec. 1**, current methods utilizing MLLM for VideoQA simply concatenate the features of uniformly sampled frames, which is not sufficient for tasks featuring causal-temporal reasoning. In light of this, we extract frame features with temporal information (**Sec. 3.1**), select features of continuous frames that are crucial for answering the question (**Sec. 3.2**), and preserve the general context of the whole video (**Sec. 3.3**). It is particularly emphasized that the core of our method, the question-critical keyframes retriever (**Sec. 3.2**), selects frames in a **hard** way, which means we didn't change the spatial-temporal features $V _{ST} \in R^{T \times N _C \times D _C}$ where we extracted in **Sec. 3.1**, and we just select W features of continuous frames in $V _{ST} \in R^{T \times N _C \times D _C}$ as our question-critical
> spatial-temporal frame features $V _{Key} \in R^{W \times N _C \times D _C}$. For comparison, the attention mechanism in [1,2,3,4] is used in a **soft** way, which means they change the frame features they extracted by the weighted summation in attention mechanism.
>
> **(3) A detailed discussion of different strategies for frame selection.**
>
> Here, we compare the mechanism of frame selection in our method with the two latest works in complex VideoQA, including TranSTR [2] and MIST [3]. The core difference between our method and others is that **we select continuous frames as the question-critical event, instead of discrete frames separately distributed across different timestamps within videos.**  For example, TranSTR [2] selects K frames according to the highest k scores in the cross-attention map between frame embeddings and question embeddings, with the perturbed maximum method [4]. MIST [3] selects K frames by simply conducting Gumbel-Softmax sampling K times. Both of them result in a selection of K **discrete** frames and would disrupt the intrinsic coherence between adjacent frames, which we think cannot constitute an event ("event" here means K **continuous** frames). Therefore, we use a **1D-Conv** with a window size of K on the attention scores $M \in R^{T}$ to get an event level scores $\hat{M} \in R^{T-K+1}$ (illustrated **in the 2nd paragraph of Sec. 3.2**), followed with a Gumbel-Softmax and a matrix multiplication with the transformation matrix we defined in **Equation (4)**. For more fair comparisons with the method using MLLM, we have added another experiment to verify the effectiveness of our proposed question-critical keyframes retriever based on the InstructBLIP-Vicuna-7B in the NExT-QA test set as shown in the following table,
> | Method      | Acc |
> | ----------- | ----------- |
> | InstructBLIP + uniform| 68.4%  |
> | InstructBLIP + MIST   | 69.3%  |
> | InstructBLIP + TranSTR| 71.1%  |
> | InstructBLIP + E-STR| 71.7%  |
> "InstructBLIP + uniform" means concatenating uniformly sampled frames without selection, and " + MIST/TranSTR" means selecting frames with the method in MIST and TranSTR as we talked about above. From the results, our selection method still achieved the highest performance compared to other methods.
>
> \
> ***References***
> \
> [1] Jiyang Gao, Runzhou Ge, Kan Chen, and Ram Nevatia. Motion-appearance co-memory networks for video question answering. In CVPR, 2018
> \
> [2] Yicong Li, Junbin Xiao, Chun Feng, Xiang Wang, and Tat-Seng Chua. Discovering spatio-temporal
> rationales for video question answering. In ICCV, 2023
> \
> [3] Difei Gao, Luowei Zhou, Lei Ji, Linchao Zhu, Yi Yang, and Mike Zheng Shou. Mist: Multi-modal
> iterative spatial-temporal transformer for long-form video question answering. In CVPR, 2023.
> \
> [4] in Jiang and Yahong Han. Reasoning with heterogeneous graph alignment for video question answering. In AAAI, 2020
> \
> [5] Quentin Berthet, Mathieu Blondel, Olivier Teboul, Marco Cuturi, Jean-Philippe Vert, and Francis Bach. Learning with differentiable pertubed optimizers. NeurIPS, 2020

---

> > ### Author Response · Authors · 2023-11-12
> >
> > **(4) Why methods like MERLOT [6,7] work reasonably well on VideoQA.**
> >
> > As we have discussed in **SUPPLEMENTARY C**, methods pre-trained on large-scale video-text data (including but not limited to MERLOT, VQA-T (Yang et al., 2021), HiTea (Ye et al., 2022), InterVideo (Wang et al., 2022)...) work reasonably well on simple VideoQA tasks (e.g., MSVD-QA, MSRVTT-QA, ActivityNet-QA) because **video-text pretraining** can substantially enhance the ability to recognize objects and attributes in videos, which is the primary focus of simple VideoQA tasks. However, the challenging skills of causal temporal reasoning, essential in complex VideoQA tasks, are the strengths of our method, which identifies question-critical video segments and eliminates redundancy to enhance reasoning. We also compare methods with large-scale video-text pre-training in **Table 1 and 11 (methods followed with ∗)**. We emphasize some results below:
> > |               | MSVD-QA | MSRVTT-QA | ActivityNet-QA |STAR|
> > | ----------- | ----------- | ----------- | ----------- | ----------- |
> > | MERLOT | - | 43.1% | 41.4% | 40.5% |
> > | HiTea | 55.6% | 45.9% | - | - |
> > | InterVideo | 55.5% | 47.1% | - | 58.7% |
> > | InstructBLIP | 59.6% | 46.9% | 45.9% | 61.2% |
> > | +E-STR   | 61.1%  | 49.1% | 49.2% | 64.7% |
> >
> > \
> > **Q2: Difficulty in understanding our paper.**
> >
> > To help you understand our method in detail, we will introduce where each symbol comes from.
> >
> > **(1) $X _{l} \in R^{T \times N _I \times D _I}$, the output of the $l$-th transformer block (with adapters in Figure 3 (a)) in vision transformer.**
> >
> > As we have mentioned in **Sec. 3.1, Para. 1, the 9th line**, the **$X _{l} \in R^{T \times N _I \times D _I}$** is the output of the $l$-th transformer block **(with adapters)** in the vision transformer, which can be obtained with **Equation (1)**. $T$ is the number of sampled frames (32 here), $N _I$ is the patch number (257 here), and $D _I$ is the embedding dimension (1408 here) of the image encoder.
> >
> > **(2) $X _{L} \in R^{T \times N _I \times D _I}$, the output of the last transformer block (with adapters) in the vision transformer.**
> >
> > As we have mentioned in **Sec. 3.1, Para. 2, the 1st line**, the **$X _{L} \in R^{T \times N _I \times D _I}$** is the output of the **last** transformer block **(with adapters)** in vision transformer. The subscript is $L$ because the number of blocks of the vision transformer is $L$.
> >
> > **(3) spatial-temporal frame features $V _{ST} \in R^{T \times N _C \times D _C}$, the output of the Qformer with $X _{L} \in R^{T \times N _I \times D _I}$ as inputs.**
> >
> > As we have mentioned in **Sec. 3.1, Para. 2, the 3rd line**, we proceed to feed $X _{L} \in R^{T \times N _I \times D _I}$ into the Qformer, and the output is our spatial-temporal frame features $V _{ST} \in R^{T \times N _C \times D _C}$, where $N _C$ is the number of visual tokens (32 here), and $D _C$ is the hidden size of the Qformer (768 here).
> >
> > **(4) spatial frame embeddings $E _{S} \in R^{T \times D _I}$, the output of the completely frozen vision transformer (without adapters).**
> >
> > As we have mentioned in **Sec. 3.1, Para. 2, the 7th line**, by feeding frames into the totally frozen vision transformer, we can obtain spatial frame embeddings $E _{S} \in R^{T \times N _I \times D _I}$. Since we only reserve the class tokens, we will obtain $E _{S} \in R^{T \times D _I}$.
> >
> > **(5) question embeddings $E _{Q} \in R^{1 \times D _I}$, the output of the frozen text encoder.**
> >
> > As we have mentioned in **Sec. 3.2, Para. 1, the 4th line**, by feeding the questions into the text encoder, we can obtain question embeddings $E _{Q} \in R^{1 \times D _I}$. The sequence length is 1 because we only reserve the class token, and its embedding dimension is the same as $E _{S} \in R^{T \times D _I}$.
> >
> > **(5.5) The difference between "embeddings" and "features" in our paper.**
> >
> > The main difference between the **embeddings $\in R^{N _I \times D _I}$** and **features $\in R^{N_C \times D _C}$** in our paper is that, **features** refers to **embeddings** through the Qformer **(features $\in R^{N_C \times D _{C}}$ = Qformer (embeddings $\in R^{N _I \times D _{I}}$))**, and this is why the dimensions of **"spatial frame embeddings" $D _I$** and  **"spatial-temporal frame features" $D _C$** not consistent.
> >
> > Besides, $V _{ST} \in R^{T \times N _C \times D _C}$ is obtained through the vision transformer **with adapters**, and the adapters learn to build the relationships between frames, resulting in the term of **"spatial-temporal features"**. For comparison, $E _{S} \in R^{T \times D _I}$ is obtained through the completely frozen vision transformer **without adapters**, and each frame is encoded independently without temporal intercation, leading to the term **"spatial embeddings"**.
> >
> >
> > ***References***
> >
> > [6] Zellers et al., MERLOT: Multimodal Neural Script Knowledge Models, in NeurIPS 2021.
> > \
> > [7] Zellers et al., MERLOT Reserve: Neural Script Knowledge through Vision and Language and Sound, in CVPR 2022.

---

> > > ### Author Response · Authors · 2023-11-12
> > >
> > > **(6)  attention scores $M \in R^{T}$, the attention map between $E _S \in R^{T \times D _I}$ and $E _Q \in R^{1 \times D _I}$.**
> > >
> > > As we have mentioned in **Sec. 3.2, Para. 2, the 6th line**,  a cross-attention layer is applied by taking the  $E _S \in R^{T \times D _I}$ as query and the $E _Q \in R^{1 \times D _I}$ as key, which yields the cross attention map $M \in R^{T}$, recording the attention scores between each frame and the question.
> > >
> > > **(7)   event-level scores $\hat{M} \in R^{T-W+1}$.**
> > >
> > > As we have mentioned in **Sec. 3.2, Para. 2, the 8th line**,  we apply one-dimensional convolution with a window size of $W$ to operate on the $M \in R^{T}$, resulting in $\hat{M} \in R^{T-W+1}$. Each entry $\hat{M} _i$ in $\hat{M}$ is computed with the sum of $W$ entries in $M$: $\hat{M} _i = \sum _{j=i} ^{i+W-1}M _j$. This operator ensures that we will choose the frames that are **continuous**.
> > >
> > > **(8)   one hot vector $\widetilde{M} \in R^{T-W+1}$.**
> > >
> > > As we have mentioned in **Sec. 3.2, Para. 5, and Equation (3)**,  we resort to the Gumbel-Softmax on $\hat{M} \in R^{T-W+1}$ to obtain the one hot vector $\widetilde{M} \in R^{T-W+1}$ for the **differentiability** of our question-critical keyframes retriever (**Argmax in Equation (2) is non differentiable**).
> > >
> > > **(9)  constant transformation matrix $H \in R^{(T-W+1) \times W \times T}$, and key event mask $Y \in R^{W \times T}$.**
> > >
> > > As we have mentioned in **Sec. 3.2, Para. 6, and Equation (4), and the bottom right of Figure 3 (c)**, the constant transformation matrix $H \in R^{(T-W+1) \times W \times T}$ helps us to get the key event mask $Y \in R^{W \times T}$ by apply matrix multiplication with ($Y \in R^{W \times T}) = (\widetilde{M} \in R^{T-W+1}) \times (H \in R^{(T-W+1) \times W \times T})$.
> > >
> > > **(10)  question-critical spatial-temporal frame features $V _{Key} \in R^{W \times N_C \times D_C}$.**
> > > As we have mentioned in **Sec. 3.2, Para. 1** that, the question-critical keyframes retriever is responsible for selecting $W$ continuous question-critical spatial-temporal frame features $V _{Key} \in R^{W \times N_C \times D_C}$ from $V _{ST} \in R^{T \times N_C \times D_C}$ in a differential way. With the key event mask $Y \in R^{W \times T}$, we can realize the goal by matrix multiplication in **Sec. 3.2, the last paragraph**: $(V _{Key} \in R^{W \times N_C \times D_C}) = (Y \in R^{W \times T}) \times (V _{ST} \in R^{T \times N_C \times D_C})$.
> > >
> > > With the question-critical spatial-temporal frame features $V _{Key} \in R^{W \times N_C \times D_C}$, we can evidently boost the performance of MLLM in complex VideoQA tasks as shown in **Table 1 and 2**.
> > >
> > >
> > > **To give you an intuitive understanding of the symbols above, we denote the key symbols that we talked about above with red bounding boxes in the following URL:**
> > >
> > > https://s2.loli.net/2023/11/12/I68i34jPqnrAJx2.png
> > >
> > > https://s2.loli.net/2023/11/12/etMBmTQlYLS4CIN.png
> > >
> > > We sincerely hope that you can have a deeper understanding of our method through our replies to your valuable comments, and we are looking forward to your further feedback.

---

> ### Comment · Reviewer_zSXn · 2023-11-13
> **Thank you for your quick response.**
>
> Dear authors.
>
> I sincerely thank you for your fast response. I carefully read the whole, and I understand what the authors mentioned. However, I still don't think the shape of the current manuscript is the most effective way of delivering the main point of this work and could lead to misunderstanding on a few parts if someone did not check this thread.
>
> First, regarding the attention-related part, the authors also agree that the attention mechanism and continuous frame selection is NOT a new approach. Prior works generally subsample every n frame (someone may say it is just a matter of changing fps unless it drops a large amount of information, e.g., 32 fps to 4 fps). However, most approaches presented before MLLM sell dynamic frame selection as more effective (with 3D conv, memory network, etc). This paper tries to show that such trends do not hold in MLLM, and I believe that is (one of) the core message of this paper. However, I think this flow is not well delivered to the current manuscript (and it is very easy to fall into misunderstanding).
>
> Also, I did check the dimensions in the paper but failed to find the detailed equations from all in the current main manuscript. As those are the critical components, they need to be very clearly delivered in the manuscript (some people may be able to 'decode' based on the current response, but it is still challenging without this response; i.e., it is hard to say the readers can replicate this approach just with the manuscript. Plus, the texts in the figures as those are extremely small). It would be even better if the author could back-reference the exact equation of each matched part in the experiments (I still think Sections 1-3 and 4-5 need to be better connected).
>
> In sum, I do think this work needs some significant revision (Sections 1-3, especially). As a reviewer who really believes it could be unfair to check multiple revisions within the review period (I do believe the author-rebuttal discussion period is mainly for resolving the misunderstandings, not for revisiting the writing multiple times), I will not add comments on the revised version. However, I will revisit the main manuscript or this thread (without any biases) during the reviewer discussion period (NOT during the author-reviewer discussion period) and revise my score based on the revision.

---

### Official Review · Reviewer_92w2 · 2023-10-30

**Soundness:** 3 good
**Presentation:** 3 good
**Contribution:** 2 fair
**Rating:** 5
**Confidence:** 5

**Summary:**

This paper introduces E-STR, which aims to handle complex VideoQA tasks involving long videos with multiple objects and events. E-STR incorporates a question-critical keyframes retriever to adaptively select key events for spatial-temporal reasoning, along with a context encoder to preserve general video context.

**Strengths:**

1) The paper proposes a reasonable method to handle the complex VideoQA task by keyframe retrieval, which can effectively compress the long video (32->6). The idea of context encoder is innovative, which seems different from other similar works.
2) A series of experiments on complex VideoQA benchmarks have demonstrated the superiority of the method.
3) The article has a clear structure, logical writing, and is easy to understand.

**Weaknesses:**

1) The retrieval-based approach is not entirely new, as many existing works [1]-[3] utilize this idea for the VideoQA task. The article lacks a detailed comparison and analysis of these works. For example, in MIST, a similar keyframes-obtaining method based on attention maps is proposed, and SeViLA[1] introduces prompting LLM to get keyframes. Why does the paper choose a 1d CNN to find keyframes, and what is its advantage? What’s more, the results seem not as good as SeViLA in the NExT-QA and STAR datasets, what about the reasons?
2) Simply adapting the InstructBLIP to VideoQA tasks already achieves relatively strong performances (63.2->69.5), thus the performance gains seem to rely on the pre-trained MLLM (69.5->72.8). Besides, the contributed GCE & ST seem to have weak performance gain in Tab. 5.
3) The paper aims to handle long video reasoning. STAR only contains videos of 12s on average. More complex benchmarks like AGQA v2 (large-scale compositional reasoning), and ActivityNet-QA (longer videos of 180s on average) are worth evaluating.
4) Need further qualitative results to prove the effectiveness of the method.
5) Limitations are not discussed.

[1] Shoubin Yu, Jaemin Cho, Prateek Yadav, and Mohit Bansal. Self-chained image-language model for video localization and question answering. arXiv preprint arXiv:2305.06988, 2023.
[2] Sungdong Kim, Jin-Hwa Kim, Jiyoung Lee, and Minjoon Seo. Semi-parametric video-grounded text generation. arXiv preprint arXiv:2301.11507, 2023.
[3] Tianwen Qian, Ran Cui, Jingjing Chen, Pai Peng, Xiaowei Guo, and Yu-Gang Jiang. Locate before answering: Answer guided question localization for video question answering. IEEE Transactions on Multimedia, 2023.

**Questions:**

1) Why remain the "spatial" frame feature for question retrieval, does it keep complete video content? What's the difference between the "spatial frame feature" and the "spatial-temporal frame feature" (ST feature)? Why are the dimensions of these two features not consistent?
2) What are the differences between the proposed ST feature adapter and the current Adapter-based works, esp. the ST-Adapter (Pan et al. 2022)?

---

> ### Author Response · Authors · 2023-11-11
>
> We greatly appreciate your constructive and insightful comments, and we address your major concerns below.
> \
> \
> **Weakness 1:  A detailed comparison and analysis of works utilizing frame selection in VideoQA.**
> \
> \
> **For method with MLLM.**
> \
> As far as we know, SeViLA [1] is currently the only work that utilizes an MLLM for VideoQA. However, there are several main differences between SeViLA and our method. **Firstly, SeViLA adopts a two-stage training, while our method is a differential selection with end-to-end training.** SeViLA firstly per-train the model on the QVHighlights for 80 epochs and then finetune the model on downstream tasks. Instead, our method only needs to be fine-tuned once on downstream tasks thanks to the differentiability of our question-critical keyframes retriever. **Secondly, SeViLA locates frames by prompting LLM, while we compute the attention map as the basis for our frame selection.** We introduce the text encoder, which has been jointly pre-trained with the image encoder, to compute the attention map between spatial frame embeddings $E _S \in R^{T \times D _{I}}$, and question embeddings $E _Q \in R^{1 \times D _{I}}$. We then select the continuous  W frames that are most crucial in a differentiable way. **Thirdly, SeViLA extracts spatial frame features via the frozen ViT independently, while we extract frame features with temporal information.** We add trainable adapters in the frozen ViT (**Figure 3 (a)**) so that we can build temporal relationships between frames during feature extraction (**Sec 3.1**).
>
> **For method without MLLM, and why we choose a 1d Conv.**
> \
> As you mentioned in your reviews, there are some other works that present frame selection in VideoQA. Here, we compare the mechanism of frame selection in our method with the two latest works in complex VideoQA, including TranSTR [2] and MIST [3]. The core difference between our method and others is that **we select continuous frames as the question-critical event, instead of discrete frames separately distributed across different timestamps within videos.**  For example, TranSTR [2] selects K frames according to the highest k scores in the cross-attention map between frame embeddings and question embeddings, with the perturbed maximum method [4]. MIST [3] selects K frames by simply conducting Gumbel-Softmax sampling K times. Both of them result in a selection of K **discrete** frames and would disrupt the intrinsic coherence between adjacent frames, which we think cannot constitute an event ("event" here means K **continuous** frames). Therefore, we use a **1D-Conv** with a window size of K on the attention scores $M \in R^{T}$ to get an event level scores $\hat{M} \in R^{T-K+1}$ (illustrated in the 2nd paragraph of **Sec. 3.2**) followed with a Gumbel-Softmax and a matrix multiplication with the template matrix we defined in **Equation (4)**. For more fair comparisons with the method using MLLM, we have added another experiment to verify the effectiveness of our proposed question-critical keyframes retriever based on the InstructBLIP-Vicuna-7B in the NExT-QA test set as shown in the following table,
> | Method      | Acc |
> | ----------- | ----------- |
> | InstructBLIP + uniform| 68.4%  |
> | InstructBLIP + MIST   | 69.3%  |
> | InstructBLIP + TranSTR| 71.1%  |
> | InstructBLIP + E-STR| 71.7%  |
> "InstructBLIP + uniform" means concatenating uniformly sampled frames without selection, and " + MIST/TranSTR" means selecting frames with the method in MIST and TranSTR as we talked about above. From the results, our selection method still achieved the highest performance compared to other methods.
>
> **Results comparison with SeViLA.**
> \
> The results on **Table 1 and 2** are from the model with Vicuna, while **Table 3** shows that models with FLAN-T5 perform much better (**74.5** on NExT-QA, surpassing the results of **73.8** in SeViLA). For a clear comparison with SeViLA, we reorganize the results of models with FLAN-T5:
> | Method  | NExT-QA| STAR|
> | ----------- | ----------- | ----------- |
> | SeViLA  | 73.8%    | 64.9%    |
> | Ours      | 74.5%    | 64.9%  |
> The results indicate that our method achieves better performance in NExT-QA **(+0.7)** and an equal Acc on STAR. Notably, videos in NExT-QA are longer than STAR.
>
>
> \
> ***References***
> \
> [1] Shoubin Yu, Jaemin Cho, Prateek Yadav, and Mohit Bansal. Self-chained image-language model for video localization and question answering. arXiv preprint arXiv:2305.06988, 2023.
> \
> [2] Yicong Li, Junbin Xiao, Chun Feng, Xiang Wang, and Tat-Seng Chua. Discovering spatio-temporal
> rationales for video question answering. In ICCV, 2023
> \
> [3] Difei Gao, Luowei Zhou, Lei Ji, Linchao Zhu, Yi Yang, and Mike Zheng Shou. Mist: Multi-modal
> iterative spatial-temporal transformer for long-form video question answering. In CVPR, 2023.
> \
> [4] Quentin Berthet, Mathieu Blondel, Olivier Teboul, Marco Cuturi, Jean-Philippe Vert, and Francis Bach. Learning with differentiable pertubed optimizers. NeurIPS, 2020

---

> ### Author Response · Authors · 2023-11-11
>
> **Weakness 2:  Performance gains seem to rely on the pre-trained MLLM.**
> \
> \
> Since our target is to improve video-language reasoning in complex VideoQA tasks on the basis of MLLM, and a powerful MLLM would be our first choice. Intuitively, a weaker MLLM will lead to a performance drop while we don't think there is a necessity for this evaluation. Instead of looking back, we believe more powerful MLLM in the future will empower our work better. Also, we would like to emphasize that the performance improvement mainly benefits from the E-STR method, composed of trainable adapters, a well-designed question-critical keyframes retriever, and a general context encoder, instead of a simple adaptation. Although our adapters and general context encoder are not as effective as the question-critical keyframes retriever, they can still promote video reasoning in complex VideoQA tasks (**(w/o ST (-0.7)), (w/o GCE (-0.5)) in Table 5**) which we think is necessary.
> \
> \
> **Weakness 3,4,5:  More benchmarks, further qualitative results, and limitations:**
> \
> \
> **More experiments on other benchmarks are in the SUPPLEMENTARY C of the original paper.**
>
> We applied our method to additional VideoQA tasks, including **MSVD-QA**, **MSRVTT-QA**, and **ActivityNet-QA**.
> |               | MSVD-QA | MSRVTT-QA | ActivityNet-QA |
> | ----------- | ----------- | ----------- | ----------- |
> | InstructBLIP | 59.6% | 46.9% | 45.9% |
> | +E-STR   | 61.1%  | 49.1% | 49.2% |
>
> **More qualitative results and further discussions about the limitations are in the SUPPLEMENTARY D of the original paper.**
>
> We selected more typical examples for both successive and failure analysis, along with the discussions about the limitations (**Sec 4.4 and Supplementary D**).
>
> \
> \
> **Question 1:  "spatial frame features (--> embeddings)" and the "spatial-temporal frame features"**
> \
> \
> **spatial frame embeddings**
>
> Thanks for your careful review of our paper, we first make a minor correction in your expression. In our paper, we use **"spatial frame embeddings"** to refer to the spatial frame features in your review. As we defined in **the 2nd paragraph, Sec.3.1**, the spatial frame embeddings **$E _S \in R^{T \times N _I \times D _I}$** are extracted from a totally frozen image encoder **without adapters** (illustrated in the upper right of Figure 3 (a), since each frame is encoded independently without temporal information, we only use the term **"spatial"**), $N _I$ is the patch number of each frame (including the class token), and $D _I$ is the embedding dimension. We only keep the class tokens of  $E _S \in R^{T \times N _I \times D _I}$, resulting in $E _S \in R^{T \times D _I}$.
>
> **spatial-temporal frame features**
>
> The main difference between the **embeddings $\in R^{N _I \times D _I}$** and **features $\in R^{N_C \times D _C}$** in our paper is that, **features** refers to **embeddings** through the Qformer **(features $\in R^{N_C \times D _{C}}$ = Qformer (embeddings $\in R^{N _I \times D _{I}}$))**, and this is why the dimensions of **"spatial frame embeddings" $D _I$** and  **"spatial-temporal frame features" $D _C$** not consistent. Similarly, we get the spatial-temporal embeddings $E _{ST} \in R^{T \times N _I \times D _I}$ by feeding frames into the image encoder **with adapters** (the adapters learn to build the relationships between frames, so we use the term **"spatial-temporal"**). We then feed $E _{ST} \in R^{T \times N _I \times D _I}$ into the Qformer to get the **"spatial-temporal features"** $V _{ST} \in R^{T \times N _C \times D _C}$ (defined in **the 2nd paragraph, Sec.3.1**).
>
> **The differences between  $E _S \in R^{T \times D _I}$ and $V _{ST} \in R^{T \times N _C \times D _C}$ and why we remain $E _S \in R^{T \times D _I}$**
>
> Since $E _S \in R^{T \times D _I}$ are extracted from the completely frozen image encoder in EVA-CLIP, they are naturally aligned with the question embeddings $E _Q \in R^{D _I}$ which we extracted with the frozen text encoder in EVA-CLIP (also supported by the ablation results in **Table 6**). The  $E _S$ and $E _Q$ are only responsible for computing the attention map $M \in R^{T}$, which will be used to select the continuous W frame features in $V _{ST} \in R^{T \times N _C \times D _C}$, resulting in the $V _{Key} \in R^{W \times N _C \times D _C}$ (as illustrated in **Sec. 3.2**).
>
> \
> \
> **Question 2:   About the ST-Adapter [1]**
> \
> \
> The original ST-Adapter is only a 3D-Conv kernel before the MHSA layer in the transformer block (**Figure 3 (a)**), we additionally introduce a parallel MLP-based Adapter to refine the frozen parameters in the MHSA and FFN layers to align better with the ST-Adapter. The ablation results in **Table 8** validate this.
>
>
> [1] Junting Pan, Ziyi Lin, Xiatian Zhu, Jing Shao, and Hongsheng Li. St-adapter: Parameter-efficient image-to-video transfer learning. In NeurIPS, 2022.

---

### Official Review · Reviewer_bWa6 · 2023-10-31

**Soundness:** 2 fair
**Presentation:** 3 good
**Contribution:** 3 good
**Rating:** 5
**Confidence:** 4

**Summary:**

This paper proposes to address the task of complex video question answering. To reduce the complexity of previous methods, this paper introduces a two-step approach. Specifically, it first designs a moment adapter to retrieve the question-related frames. Then, it associates corresponding critical information with the general contexts of the unselected part to predict the answer. Besides, it also incorporates lightweight adapters within the frozen image encoder. Experiments are conducted on three datasets.

**Strengths:**

1. The motivation of this paper is straightforward and easy to follow.
2. This paper is well-written and easy to read.
3. Supplementary file is provided.

**Weaknesses:**

1. The novelty is limited. This paper proposes a two-step approach which first retrieves the question-related moment and then achieves reasoning. This process is similar to the coarse-to-fine approach in many temporal grounding methods, for example, but not limited to, “Scanning Only Once: An End-to-end Framework for Fast Temporal Grounding in Long Videos”. Since the motivation is straightforward, the newly introduced technical designs are not new and not exciting. Therefore, I believe that the novelty is incremental.

2. Missing some relevant references. Since the main approach is coarse-to-fine, the authors should add and compare more related methods to discuss their differences.

3. Experiments are not fair. This paper proposes a two-step approach, directly comparing it with other one-step approaches is unfair. Although this work brings large improvements, it also leads to higher running time and GPU cost. Therefore, the authors should re-implement other two-step approaches from other tasks into the current task for comparison.

4. The efficiency comparison in Table 4 is not convincing. In general, a two-step approach will cost much time and GPU memory. The authors should provide a detailed analysis of each component of the proposed method to demonstrate its efficiency.

**Questions:**

Please see the weakness.

---

> ### Author Response · Authors · 2023-11-11
>
> We greatly appreciate your constructive and insightful comments. Please allow us to restate the novelty of our work and the differences from other works.
> \
> \
> **Weakness 1 and 2: limited novelty and differences from other works**
> \
> \
> Different from the annotations in the datasets of Temporal Grounding, which explicitly annotate the timestamps for which segments in videos correspond to which query, the annotations in VideoQA only involve the tuple of $<question, answer, video>$. In light of this, we can only learn to seek the question-critical moment in VideoQA without explicit supervision, and the methods in the task of Temporal Grounding are not suitable here (e.g., the boundary regression loss $L _{reg}$ in [1] and the moment localization loss $L _{moment}$ in [2] are not applicable here). Therefore, the newly introduced technical designs lie in the question-critical keyframes retriever in **Sec. 3,2**. For further explanation, we compare the mechanism of frame selection in our method with the two latest works in complex VideoQA, including TranSTR [3] and MIST [4].
> \
> \
> **(1) We select continuous frames as the question-critical event, instead of discrete frames separately distributed across different timestamps within videos.**  TranSTR [3] selects K frames according to the highest k scores in the cross-attention map between frame embeddings and question embeddings, with the perturbed maximum method [5]. MIST [4] selects K frames by simply conducting Gumbel-Softmax sampling K times. Both of them result in a selection of K **discrete** frames and would disrupt the intrinsic coherence between adjacent frames, which we think cannot constitute an event ("event" here means K **continuous** frames). Therefore, we use a 1D-Conv with a window size of K on the attention scores $M \in R^{T}$ to get an event level scores $\hat{M} \in R^{T-K+1}$ (illustrated in the 2nd paragraph of **Sec. 3.2**) followed with a Gumbel-Softmax and a matrix multiplication with the template matrix we defined in **Equation (4)**. Further, we fully utilize both the text encoder and the image encoder in CLIP. TranSTR encodes the questions with DeBERTa, and MIST encodes the questions with BERT. However, as the studies shown in **Table 6**, only the text encoder which has undergone large-scale contrastive learning with the image encoder can yield the highest performance since their shared understanding of the visual and textual content.
> \
> \
> **(2) We introduce learnable adapters to learn the temporal relationships between frames better** As illustrated in **Sec. 3.1**, the vanilla ViT in CLIP cannot capture the temporal relationships and we introduce a 3D-Conv-based ST-Adapter [6] together with a parallel MLP-based adapter (in **Figure 3 (a)**) to make the process of feature extraction temporal aware. Instead, TranSTR and MIST extract the features of each frame independently, resulting in a lack of temporal relationships between frames. Our ablation studies in **Table 5 (w/o ST (-0.7))** and **Table 8** support the effectiveness of our method.
>
> \
> ***References***
> \
> [1] Yulin Pan and Xiangteng He and Biao Gong and Yiliang Lv and Yujun Shen and Yuxin Peng and Deli Zhao. Scanning Only Once: An End-to-end Framework for Fast Temporal Grounding in Long Videos. In ICCV, 2023.
> \
> [2] Jie Lei, Tamara L Berg, and Mohit Bansal. Qvhighlights: Detecting moments and highlights in videos via natural language queries. In NeurIPS, 2021.
> \
> [3] Yicong Li, Junbin Xiao, Chun Feng, Xiang Wang, and Tat-Seng Chua. Discovering spatio-temporal
> rationales for video question answering. In ICCV, 2023
> \
> [4] Difei Gao, Luowei Zhou, Lei Ji, Linchao Zhu, Yi Yang, and Mike Zheng Shou. Mist: Multi-modal
> iterative spatial-temporal transformer for long-form video question answering. In CVPR, 2023.
> \
> [5] Quentin Berthet, Mathieu Blondel, Olivier Teboul, Marco Cuturi, Jean-Philippe Vert, and Francis Bach. Learning with differentiable pertubed optimizers. NeurIPS, 2020
> \
> [6] Junting Pan, Ziyi Lin, Xiatian Zhu, Jing Shao, and Hongsheng Li. St-adapter: Parameter-efficient image-to-video transfer learning. In NeurIPS, 2022.

---

> ### Author Response · Authors · 2023-11-11
>
> **Weakness 3: Experiments are not fair.**
> \
> \
> From your constructive reviews, we think the "two-stage" means an additional involvement in the process of frame selection. In our main results of **Table 1** and **Table 2**, the methods of MIST and TranSTR all involve a process of frame selection as we talked about above. **Based on your valuable comment, for more fair comparisons with the method using MLLM**, we have added another experiment to verify the effectiveness of our proposed question-critical keyframes retriever based on the InstructBLIP-Vicuna-7B in the NExT-QA test set as shown in the following table,
> | Method      | Acc |
> | ----------- | ----------- |
> | InstructBLIP + uniform| 68.4%  |
> | InstructBLIP + MIST   | 69.3%  |
> | InstructBLIP + TranSTR| 71.1%  |
> | InstructBLIP + E-STR| 71.7%  |
> "InstructBLIP + uniform" means concatenating 5 uniformly sampled frames without selection, and " + MIST/TranSTR" means selecting 5 frames with the method in MIST and TranSTR as we talked about above. From the results, our selection method still achieved the highest performance compared to other methods.
>
> **Weakness 4: Efficiency comparison.**
> \
> \
> During inference, the only added parameters are all lightweight. For example, the parameters of our adapters, question-critical keyframes retriever, and general context encoder are 28M, 2M, and 9M respectively. Since the majority of computation costs of the MLLM come from the LLM, we can achieve a fair computation efficiency as well as reduce the tokens directly fed into the LLM, that's also what our methods do as we mentioned in *Sec. 4. Computation efficiency.*. In **Table 4**, our method achieves a GFlops of **9673**, which is close the to GFlops of **9382** of **concat-6**, because we only feed **6x32** visual tokens into the LLM (the same as **concat-6**). Instead, concat-32 has a GFlops of **15616** because it feeds **32x32** visual tokens into the LLM. We display more results of the computation of efficiency below:
> | Method | Visual tokens length | Latency (B=8) | Latency (B=16) | Latency (B=32) | GFlops (B=8) | GFlops (B=16) | GFlops (B=32) | GPU Mem (B=16) | Throughout (B=16) |
> | ----------- | ----------- | ----------- | ----------- | ----------- | ----------- | ----------- | ----------- | ----------- | ----------- |
> | Concat-6   | 6x32    | 1606ms | 2974ms| 5698ms| 81524ms   | 163421    | 327125| 25.14GB    | 5.32V/s|
> | Concat-12 | 12x32  | 1666ms | 3074ms| 5915ms| 91657ms   | 183628    | 368171| 28.08GB| 4.85V/s|
> | Concat-24 | 24x32  | 1835ms | 3312ms| 6736ms| 111958ms | 224353| 449326    | 34.85GB| 4.03V/s    |
> | Concat-32 | 32x32  | 1966ms | 3607ms| 7237ms| 125513ms| 251438| 503577| 44.21GB    | 3.51V/s|
> | ours           | 6x32    | 1729ms | 3429ms| 6392ms| 84867ms  | 169836    | 340364| 26.41GB| 4.72V/s|
>
> We compare more latency time and GFlops with different batch sizes during inference, along with the GPU memory and throughput with a batch size = 16, on a single NVIDIA A100 GPU. We still utilize NExT-QA and InstructBLIP-Vicuna 7B for this part. The results that are summarized above show that our model performs slightly slower than Concat-6 only, **indicating that just a small overhead is introduced in inference speed by the adapters and additional modules in E-STR**.

---

### Official Review · Reviewer_xUFv · 2023-11-01

**Soundness:** 3 good
**Presentation:** 3 good
**Contribution:** 2 fair
**Rating:** 5
**Confidence:** 5

**Summary:**

Existing Multimodal Large Language Models (MLLM) still suffer from complex video question answering (VideoQA) tasks. Currently, they typically uniformly sample sparse frames and simply concatenate them to represent the entire video. However, as long and complex videos typically contain multiple events, the sparse frame sampling strategy may lead to a deficiency of essential information. To this end, they propose an event-aware spatial-temporal reasoning method E-STR. It retrieves the question-critical event before feeding the visual features into the frozen LLMs.

**Strengths:**

+ The motivation is very clear and natural. Meanwhile, the proposed method is also very straightforward.

**Weaknesses:**

+ Although the proposed method can improve the performance of baseline InstructBLIP, it is still hard to demonstrate the results are same as the initial motivation. For example, the sampled events are really important ones.

+ The main contribution of this paper is proposing an event-aware spatial-temporal reasoning strategy for VideoQA. It is still unclear how the proposed framework (cf. Figure 3) can realize "event-aware" reasoning.

**Questions:**

Based on the results in Table 4, the simple concat-32 baseline already achieves 71.1 in @All metric, which already beat all the listed state-of-the-art baselines in Table 1 (InstructBLIP with 69.5). It would be better to have more explanations about the results? Otherwise, it seems that the compared baselines are not strong enough.

---

> ### Author Response · Authors · 2023-11-11
>
> We sincerely thank you for the constructive comments. We address your major concerns below.
>
> **The relationships between the results and our motivation.**
> \
> The majority of our motivation is two folds: **(1)** build temporal relationships between frames which will be input to the frozen ViT in EVA-CLIP. **(2)** Retrieve keyframes that are crucial to answer the given question.
> \
> For motivation (1), we insert the 3D-Conv based ST-Adapter [1] into the frozen ViT in EVA-CLIP, according the conclusion in paper [1] demonstrated that “ST-Adapter is capable of extracting and leveraging the pre-trained knowledge of a large image model to achieve superior video understanding at a small parameter cost.”, which is also consistent with our ablation experimental results in Table 5 **(w/o ST (-0.7))** and Table 8.
> \
> For motivation (2), in addition to the ablation studies in Table 5 **(w/o QKR (-2.1))** which suggest the necessity of our question-critical keyframes retriever, we also show more qualitative results in Figure 7 to visualize the retrieved frames. We can observe that the retrieved frames are evidently relevant to the given questions. Although we admit that it’s hard to present a quantitative analysis (in addition to performance improvements in Table 1 and 2) to figure out whether our retriever truly discovers the question-critical moments, we additionally add another experiment to validate this.
> \
> Specifically, we transform the annotations of questions and answers in the test set of NExT-QA into a declarative sentence (e.g., **[“what is the girl doing with the bottle?”, “feeding the doll”] --> “The girl is feeding the doll”**), and we use the CLIP (ViT-large-224) to compute the similarities between the declarative ground-truth and the sampled 32 frames in videos. Here, we approximate the frame with the highest similarity as the most important among all 32 frames and determine the rate whether the frame with the highest score is among the 6 frames we have retrieved.
> | Method      | Rate |
> | ----------- | ----------- |
> | Uniformly      | 19.6%       |
> | Question-critical Keyframes Retriever   | 77.1%  |
>
> From the results, we can find that our method has a rate of 77.1% to include the frame with the highest score. For comparison, the uniformly sampled frames only have a rate of 19.6%. We believe that these conclusions can reflect the relationships between our experimental results and our motivation.
>
> **The realization of “event-aware."**
> \
> Here, we define the event as W **continuous** frames (W=5 here) and the core of our idea is to seek the continuous W frames that are most crucial among the sampled T frames (T=32 here). Figure 3 (c) demonstrates our question-critical keyframes retriever, the most important component in our framework. Briefly, given the T frame embeddings $E _S \in R^{T \times D _{I}}$, and question embeddings $E _Q \in R^{1 \times D _{I}}$, our retriever can locate the continuous W frames that are most crucial for the question according to the similarities $M \in R^{T}$ between $E _S$ and $E _Q$, and we will use the spatial-temporal features of retrieved frames for further reasoning, resulting in the term of **“event-aware”**.
>
> **Results in of concat-32 in Table 4.**
> \
> The baseline results in Table 1 and 2 are the results from InstructBLIP (Vicuna-**13B**), and the results in Table 4 come from InstructBLIP (Vicuna-**7B**) as we have mentioned in **Sec. 4.2. “Comparison with State-of-the-arts.”** that all our further experiments are conducted with InstructBLIP (Vicuna-**7B**). Therefore, the results of **concat-6** in Table 4 are consistent with the results of Vicuna-7B in Table 3 (**68.4**).
> \
> The baseline of concat-32 in Table 4 is a concatenation of all sampled 32 frames without selection, and it’s natural that its performance is better than concat-6, 12, 24 since it involves more video information. However, this result comes at the cost of an extreme length of visual tokens **(32×32)** and greater computational complexity **(15616 GFLOPs)**. Instead, our method only concatenates visual tokens of **5** selected frames along with a context representation, resulting in a smaller number of visual tokens (**(5+1)x32**, 5 is the selected frames, 1 is the general context), which is equal to the number of visual tokens in concat-6 (**6x32**, meaning the same capacity of visual information directly fed into the language model as our method) and that’s the reason why we choose the models with a concatenation of 6 uniformly sampled frames as our standard baselines in Table 1 and 2. Notably, despite concat-32 (32x32 tokens, 15616 GFLOPs) in Table 4 achieving an accuracy of 71.1, it still lags behind our method (**71.7**, with only 6x32 tokens and 9673 GFLOPs), emphasizing the effectiveness of our E-STR.
>
> ***References***
> [1] Junting Pan, Ziyi Lin, Xiatian Zhu, Jing Shao, and Hongsheng Li. St-adapter: Parameter-efficient image-to-video transfer learning. In NeurIPS, 2022.

---

### Meta-Review · Area_Chair_yXNZ · 2023-12-14

**Metareview:**

Based on the identified issues from reviewers and limited potential for improvement within the review period, I recommend rejection.

The paper proposes a two-step approach (keyframe retrieval + reasoning) similar to existing coarse-to-fine methods. Technical designs lack originality and kinda incremental. Also, as identified, comparing a two-step method to one-step methods is not really fair indeed. The performance gains seem to rely heavily on the pre-trained MLLM, and the contributed GCE & ST components have weak individual impact. Need more qualitative results to prove the method's effectiveness.

I think significant revisions are required to address the concerns raised, particularly regarding novelty, fairness of comparisons, information completeness, and clarity of writing.

**Justification For Why Not Higher Score:**

Reviewers shared concerns highlight the critical need for clear explanations of key concepts and detailed equations in the paper. The current manuscript structure and flow may lead to misunderstandings about the paper's main points, particularly regarding attention mechanisms and frame selection.

The paper could also benefit from considering more complex benchmarks like ActivityNet-QA with longer videos to demonstrate its effectiveness in handling longer video reasoning tasks, as suggested by one reviewer.

**Justification For Why Not Lower Score:**

N/A

---

### Decision · Program_Chairs · 2024-01-16

Reject